



**Enhanced Sulfate Formation in Mixed Biomass Burning and Sea-salt**
**Particles Mediated by Photosensitization: Effects of Chloride and**
**Nitrogen-containing Compounds**
Rongzhi Tang[1,2], Jialiang Ma[3], Ruifeng Zhang[4], Weizhen Cui[1], Yuanyuan Qin[5], Yangxi Chu[6],
Yiming Qin[1], Alexander L. Vogel[3], Chak K. Chan[4,*]
[1] School of Energy and Environment, City University of Hong Kong, Hong Kong, China
[2] Shenzhen Research Institute, City University of Hong Kong, Shenzhen 518057, China
[3] Institute for Atmospheric and Environmental Sciences, Goethe-University Frankfurt, 60438
Frankfurt am Main, Germany
[4] Division of Physical Science and Engineering, King Abdullah University of Science and
Technology (KAUST), Thuwal 23955-6900, Kingdom of Saudi Arabia
[5] College of Resources and Environment, University of Chinese Academy of Sciences, Beijing,
100049, China
[6] State Key Laboratory of Environmental Criteria and Risk Assessment, Chinese Research
Academy of Environmental Sciences, Beijing, 100012, China
*Correspondence to*: Chak K. Chan (chak.chan@kaust.edu.sa)
**Abstract**
Recent research has suggested that photosensitized oxidation can be an effective pathway for
the oxidation of $SO_2$ based on a limited number of model photosensitizers. However, there is a
notable dearth of research conducted on complex chemical systems, impeding a comprehensive
understanding of sulfate formation in photosensitization. This work studied sulfate formation
by mixing real biomass burning (BB) extracts and NaCl, mimicking internal mixtures of BB
and sea-salt particles. Significant enhancement of sulfate formation was observed for BB-NaCl
particles compared to incense burning (IS)-NaCl particles. For fresh particles, the sulfate
formation rate followed the trend of corn straw (CS)-NaCl>rice straw (RS)-NaCl>wheat straw
(WS)-NaCl>IS-NaCl. Aged particles were produced by irradiating the filters directly with UV
lights. Aged particles showed changes in sulfate formation rates, with the highest enhancement
by RS-NaCl due to interactions between RS and NaCl. Model experiments spiked with
nitrogen-containing organic compounds (NOCs), such as pyrazine (CHN) and 4-nitrocatechol
(CHON), revealed positive effects of chloride in the PS-CHON system and negative effects in
the PS-CHN system. Our work suggests that BB reaching or near coastal areas could affect
sulfate formation via photosensitizer-mediated reactions, potentially exacerbating air quality
concerns.
**Keywords:** sulfate formation, biomass burning, photosensitization, sea-salt aerosol, chloride



## 1 Introduction

Sulfate is a critical constituent of atmospheric particulate matter, exerting substantial influence on atmospheric radiative forcing, air quality, and human health (Fuzzi et al., 2015; Nel, 2005; Charlson et al., 1992). The commonly recognized sulfate formation mechanisms include gas-phase $SO_2$ oxidation by OH radicals (Stockwell and Calvert, 1983) and stabilized Criegee intermediates (sCIs) (Mauldin Iii et al., 2012) and multiphase and heterogeneous $SO_2$ oxidation by $H_2O_2$, $O_3$, $NO_2$, organic peroxides and $O_2$ catalyzed transition metal ions (TMI) (Seinfeld and Pandis, 2016; Wang et al., 2020a; Liu and Abbatt, 2021; Liu et al., 2020; Wang et al., 2021). More recently, some new sulfate formation pathways, e.g., in-particle nitrate photolysis (Gen et al., 2019b, a), triplet $SO_2$ chemistry (Donaldson et al., 2016; Gong et al., 2022), $SO_2$ oxidation on acidic microdroplets (Hung and Hoffmann, 2015), photosensitizer-mediated $SO_2$ oxidation (Tang et al., 2023; Wang et al., 2020b; Liang et al., 2022; Zhou et al., 2023; Wang et al., 2024b), chlorine photoactivation (Cao et al., 2024), and enhanced chlorine and photosensitization chemistry (Zhang and Chan, 2024) have been proposed. Despite extensive investigations into sulfate formation mechanisms, a substantial disparity persists between modeled simulations and measured sulfate concentrations, especially in marine boundary layer (Wyant et al., 2015) and anthropogenic emission dominated (Wang et al., 2014), highlighting the importance to further study the sulfate formation mechanism in these areas.

Biomass burning (BB) emits around 34-41 Tg of smoke aerosol annually, making it a significant contributor to both gaseous and particulate pollutants like $SO_2$, primary organic aerosol (POA), black carbon (BC) and brown carbon (BrC) (Schill et al., 2020; Laskin et al., 2015; Lin et al., 2016; Huang et al., 2022b). The recent fire outbreaks in areas like Canada, Amazonia, and Southeast Australia, together with the increased fire frequency and intensity reports in areas like western US have highlighted the risks of fire, especially BB, to human and animal health and climate change (Bond et al., 2013; Andreae, 2019; Jones et al., 2022). As an agricultural powerhouse, China boasts immense agricultural crop yields, especially in rice, wheat, and corn throughout the country. These crop residues are frequently burned in rural areas for cooking and heating purposes, as well as for land preparation after harvest, resulting in the substantial production of BrC (Chen et al., 2017). Atmospheric processes, e.g., atmospheric aging or long-range transport, can alter the chemical compositions and optical properties of BrC, potentially affecting the global climate. Recent studies have reported that the BrC species from biomass burning, e.g., vanillin (VL), acetovanillone, syringaldehyde (SyrAld) can act as photosensitizers and oxidize $SO_2$ to sulfate (Zhou et al., 2023).

Sea-salt aerosol, with its high particulate matter loadings and extensive surface area, is a crucial atmospheric constituent that plays a significant role in interfacial and multiphase reactions with reactive gases, thereby impacting global radiation balance and air quality in marine and coastal areas (Gantt and Meskhidze, 2013; Chi et al., 2015). Previous studies have observed high sulfate concentrations and light absorption properties in coastal regions when air masses passed through inland areas due to intensive biomass burning or other anthropogenic emissions, suggesting the possible interactions between the sea-salt aerosol (primarily sodium chloride) and anthropogenic emissions e.g., biomass burning (Qiu et al., 2019; Huang et al., 2018; Wu et al., 2022). For example, Qiu et al. (2019) discovered high absorption Ångström exponent (AAE



of 1.46) in coastal city Xiamen, when the air masses passing through Southeast Asia with
intense biomass burning. Van Pinxteren et al. (2015) observed an increase in sulfate
concentration (2.26 μg m$^{-3}$) during the RV MARIA S cruise as it approached the African
mainland, in contrast to the marine-origin aerosol (1.59 μg m$^{-3}$), showing significant influence
of biomass burning. Prior research has identified several secondary sulfate formation pathways
in sea-salt aerosol, e.g., multiphase $SO_2$ oxidation by $O_3$ (Alexander et al., 2012), coexistence
of $NO_2$ (Zhang and Chan, 2023), photosensitizers (Tang et al., 2023), chlorine-photosensitizer
synergistic effects (Zhang and Chan, 2024), and Cl and OH radicals generated by chlorine
photoactivation (Cao et al., 2024), highlighting the importance of NaCl-based photochemistry
in sulfate formation. Our prior study observed higher sulfate formation for incense burning-
NaCl particles than pure NaCl particles (Tang et al., 2023). The follow-up research found
magnitudes higher sulfate formation rate ($\sim$132 μM s$^{-1}$) in premixed $NH_4Cl$+IC (imidazole-2-
carboxaldehyde, a model photosensitizer found in secondary organic aerosol) particles than
pure $NH_4Cl$ particles ($\sim$ 1.8 μM s$^{-1}$) (Zhang and Chan, 2024). However, the studies on
interactions of anthropogenic emission and sea-alt aerosol on sulfate formation are very scarce.
In this study, we performed in-situ droplet experiments using BB extracts-NaCl mixture to
explore the possible interplay between biomass burning and marine aerosols in coastal areas.
BB was derived from the burning of rice straw (RS), wheat straw (WS), and corn straw (CS) as
well as incense burning (IS). This is supplemented by aqueous reactions using BB extracts and
bisulfite to mimic the in-cloud aqueous reactions of biomass burning emission-mediated S(IV)
oxidation. The effects of chloride on sulfate formation were also studied. The aims of this study
are to: (i) compare the differences in sulfate formation among different kinds of BB-NaCl
particles; (ii) evaluate the atmospheric aging (UV aging) on sulfate formation across different
BB-NaCl particles; (iii) Investigating the role of chloride ions in BB extracts mediated sulfate
formation.
**2 Material and methods**
**2.1 Burning experiments**
Three types of commonly used biomass (RS, WS and CS) were cut into small, uniform pieces
($\sim$10 cm in length) and dried. About 100 g of the dried biomass materials was then introduced
into a traditionally iron stove commonly used in rural areas. The stove was covered with a hood
and the biomass was ignited using a propane lighter. The generated BB smoke was collected
onto 90-mm quartz filters at 0.9 m$^3$ min$^{-1}$ for 10 minutes by a custom-made aerosol sampler
under mixed combustion condition (include flaming and smoldering, modified combustion
efficiency MCE, $0.85 \leq \Delta[CO_2]/(\Delta[CO_2]+\Delta[CO]) \leq 0.95$) (Ting et al., 2018). The sampler was
placed at a height of 1 meter above the ground and connected to a PM$_{2.5}$ sampling head through
a sampling pump. For incense burning (IS), laboratory generated smoldering smoke was
collected on 47-mm quartz filters at a flow rate of $\sim$ 6.0 L min$^{-1}$ for 80 min using a stainless-
steel combustion chamber. Note that the different combustion modes of IS and BB are
intentionally used to represent the real-world combustion conditions. Our previous study has
demonstrated the similarities (especially in sugars such as levoglucosan and phenols) in
GC×GC chromatograms between BB and IS (Tang et al., 2023). Hereafter, we will use BB to
represent both the real BB materials and the surrogate materials (IS) unless otherwise specified.





After sampling, the collected BB samples (fresh BB) were wrapped by pre-baked aluminum foil (550 °C for 6 h) and stored at -20 °C until further analysis.

To achieve atmospheric aging, the collected fresh BB filter samples were put into the pre-flushed (zero air, more than 24 h) combustion chamber and illuminated under UV lamps for 40 min. We used lamps of 185 nm and 254 nm, the combination of which have been widely used in oxidation flow reactor design and experiments (Peng and Jimenez, 2020; Rowe et al., 2020; Tkacik et al., 2014; Hu et al., 2022). The estimated OH exposure was ~$2.0 \times 10^{12}$ molecules $cm^{-3}$ s, equivalent to an atmospheric aging period of 15 days (assuming an average atmospheric OH concentration of $1.5 \times 10^{6}$ molecules $cm^{-3}$) (Mao et al., 2009). Detailed characterization of the OH exposure can be found in our previous study (Tang et al., 2023).

**2.2 Materials and instrumentation**

Aqueous stock solutions of BB samples were prepared by dissolving the collected filters in ultrapure water and subjecting them to ultrasonication in a cooled-water bath three times, each for 20 minutes. The resulting water extracts of the BB were then filtered through 0.22 μm PTFE filters and stored in brown vials at 4°C in a refrigerator. The anions, i.e., chloride, sulfate and nitrate of the BB extracts were analyzed by Dionex ion chromatography (ICS 1100, CA). An aliquot (~0.5 ml) of the BB or IS extracts were used for water-soluble organics detection by ultra-high performance liquid chromatography (Thermo Scientific Dionex UltiMate 3000 UHPLC) coupled with high-resolution Orbitrap Fusion Lumos Tribrid mass spectrometry (Orbitrap HRMS, Thermo Fisher Scientific, USA). The particulate organic matter was also characterized by a thermal desorption module (TDS3, Gerstel) coupled to comprehensive two-dimensional gas chromatography-mass spectrometer (GCMS-TQ™8050 NX, Shimadzu, Japan). UV-Vis spectrometry (UV-3600, Shimadzu, Japan) was employed to examine the absorbance of BB extracts. Total organic carbon (TOC) was measured by total carbon analyzer (TOC-L CPH, Shimadzu, Japan). Metal concentrations were measured by inductively coupled plasma-mass spectrometry (ICP-MS, Agilent 7800). Detailed analysis can be found in Text S1. Aqueous stock solution of sodium chloride (≥99.8%, Unichem) was prepared by dissolving the corresponding salt in ultrapure water to obtain a concentration of 1M. The study utilized high purity grade synthetic air and nitrogen supplied by the Linde HKO Ltd., while sulfur dioxide was obtained from the Scientific Gas Engineering Co., Ltd.

**2.3 Multiphase and aqueous-phase reactions of S(IV)**

For $SO_2$ uptake experiments, the stock solution of BB extracts was premixed with sodium chloride solution (1M) at a volume ratio of 1:1. A droplet generator (Model 201, Uni-Photon Inc.) was then utilized to generate droplets, which were subsequently deposited onto a hydrophobic substrate (model 5793, YSI Inc.) for $SO_2$ uptake experiments. Reactive $SO_2$ uptake experiments were performed via a flow cell/in-situ Raman system. The top and bottom quartz windows of the flow cell were used for Raman analysis and UV irradiation, respectively. The light experiment was performed using a xenon lamp (model 6258, ozone free, 300W, Newport), with photon flux of $9.8 \times 10^{15}$ photons $cm^{-2}$ $s^{-1}$ in 280-420 nm received by particles in the flow cell (Zhang and Chan, 2023). The relative humidity (RH) inside the flow cell was adjusted to 80% by mixing dry and wet synthetic air or nitrogen. The particles were then equilibrated at



80% RH for over 60 min and remained liquid throughout the experiment period. SO₂ was
introduced into the system to reach a concentration of 8.0 ppm. The prescribed size used in our
in-situ Raman research was 60 ± 5 μm. Despite using particles for droplet experiments that
were larger than ambient fine particles, we employed the SO₂ uptake coefficient ($\gamma_{SO_2}$) as a
kinetic parameter to account for the particle size effects. Comprehensive calculation of $\gamma_{SO2}$
can be found in our previous studies (Gen et al., 2019a, b; Tang et al., 2023; Zhang et al., 2020).
Aqueous-phase photochemical reactions were performed using a custom-built quartz photo
reactor (Mabato et al., 2023; 2022). Specifically, a 500 mL solution containing 100 ppm
bisulfite and 1 ppm BB TOC extracts were continuously mixed using a magnetic stirrer
throughout the experiments. To achieve air-saturated conditions, synthetic air was continuously
introduced to the solutions at a flow rate 0.5 L min⁻¹ throughout the experiments. The above
mixed solutions were then exposed to radiation via the same xenon lamp as in the droplet
experiments. Samples were collected at 1hr interval for a total of 8 h for sulfate and bisulfite
analysis using ion chromatography.

**3 Results and Discussion**
**3.1 Enhanced sulfate production of BB-NaCl droplets compared to IS-NaCl droplets.**
Figure 1 depicts the sulfate production by (a) fresh BB-NaCl; (b) aged BB-NaCl droplets as a
function of time in the presence of light, air and SO₂ at 80% RH. As our previous study (Tang
et al., 2023) has found significantly higher sulfate formation of IS-NaCl droplets over NaCl
droplets, here we only focus on the comparison of sulfate formation between different kinds of
BB-NaCl droplets and IS-NaCl droplets. Note that sulfate concentration was normalized to the
initial TOC concentration of the mixture to facilitate the comparison of sulfate production of
different droplet compositions. Regardless of whether the extracts were fresh or aged, the
sulfate production by real BB-NaCl droplets was higher than IS-NaCl droplets. Specifically,
sulfate formed by fresh (F) BB-NaCl droplets followed the trends of CS$_F$-NaCl (16.8 ± 2.6 mM
ppmC⁻¹) >RS$_F$-NaCl (9.8 ± 0.1mM ppmC⁻¹) >WS$_F$-NaCl (4.2 ± 0.2mM ppmC⁻¹) >IS$_F$-NaCl (0.8
mM ppmC⁻¹) after illumination for 1080 min. In aged (A) samples, while BB$_A$-NaCl is more
efficient than IS$_A$-NaCl in sulfate formation, the order of sulfate formation was different from
the fresh samples: RS$_A$-NaCl (35.2 ± 0.6 mM ppmC⁻¹) > CS$_A$-NaCl (13.0 ± 0.1 mM ppmC⁻¹) >
WS$_A$-NaCl (6.0 ± 1.6 mM ppmC⁻¹) > IS$_A$-NaCl (0.6 mM ppmC⁻¹). The sulfate enhancement
factors of RS$_F$-NaCl, WS$_F$-NaCl, and CS$_F$-NaCl over IS$_F$-NaCl after 18 h SO₂ uptake
(Sulfate$_{BB_F-NaCl/IS_F-NaCl}$) were 11.7, 5.0 and 20.0, respectively. The enhancement of sulfate
can also be observed in aged BB samples, with values of 54.3, 9.2 and 20.1 for RS$_A$-NaCl,
WS$_A$-NaCl, and CS$_A$-NaCl, respectively. The lower sulfate formation of IS-NaCl droplets than
BB-NaCl droplets can be explained by the significantly higher TOC concentration of IS due to
the incomplete and smoldering combustion (Table S1). The TOC concentration of the IS
extracts (>550 mg L⁻¹) was nearly an order of magnitude higher than that of the BB extracts
(34.0-69.9 mg L⁻¹), while WSOC/(WSOC+∑anions) exhibited a more than tenfold increase in
BB extracts than in IS extracts. Previous studies have confirmed that the smoldering condition
of BB will result in significantly more organic compounds and less ions than flaming condition





(Wang et al., 2020c; Fushimi et al., 2017; Kalogridis et al., 2018; Kim et al., 2018). Additionally,
significantly higher PAHs proportion (12.2%-16.6% by intensity) than IS (~5.0%) were
observed by GC×GC-MS. Huang et al. (2022a) reported higher polycyclic aromatic
hydrocarbons (PAHs) in BB particulates (CS, WS, RS, >262.5 mg kg$^{-1}$, >3.7% of organic matter)
than in IS particulates (3.3 mg kg$^{-1}$, 0.9% of organic matter) (Song et al., 2023). Fushimi et al.
(2017) and Kim et al. (2021) demonstrated that more PAHs would be emitted under flaming
compared to smoldering conditions. PAHs like pyrene, fluoranthene, and phenanthrene have
been recognized as photosensitizers (Jiang et al., 2021; Yang et al., 2021) and are mainly from
combustion processes, e.g., pyrosynthesis from aliphatic and aromatic precursors in biomass
burning processes and the constituents vary with temperatures and oxygen contents (Pozzoli et
al., 2004). The higher percentage of PAHs in BB together with the collection procedure (mixed
combustion and higher temperature for real BB while smoldering and lower temperature for IS)
suggested the BB materials would generate more PAHs at high temperatures and may contribute
to sulfate formation.
Table 1 presents the reactive ($\gamma_{SO_2}$) and normalized reactive $SO_2$ uptake coefficients ($n\gamma_{SO_2}$) of
different BB-NaCl droplets. Higher $n\gamma_{SO_2}$ were found for fresh and aged real BB-NaCl than
IS-NaCl droplets, following the trend of :CS$_F$-NaCl (8.8 × 10$^{-8}$ ppmC$^{-1}$)>RS$_F$-NaCl (6.2 × 10$^{-8}$
ppmC$^{-1}$)>WS$_F$-NaCl (2.0 × 10$^{-8}$ ppmC$^{-1}$)>IS$_F$-NaCl (0.61 × 10$^{-8}$ ppmC$^{-1}$) and RS$_A$-NaCl (2.2 ×
10$^{-7}$ ppmC$^{-1}$)>CS$_A$-NaCl (6.2 × 10$^{-8}$ ppmC$^{-1}$)>WS$_A$-NaCl (3.5 × 10$^{-8}$ ppmC$^{-1}$)>IS$_A$-NaCl (0.46
× 10$^{-8}$ ppmC$^{-1}$), respectively.
In our previous study, we observed a significant increase in sulfate formation for IS-NaCl
droplets than NaCl droplets, which we attributed to photosensitization (Tang et al., 2023).
Considering the fact that BB-NaCl droplets produced sulfate more efficiently than IS-NaCl
droplets and NaCl droplets, we explore the underlying mechanisms driving this phenomenon.
Possible reasons include nitrate (from BB extracts or newly formed) photolysis, [Cl$^-$-H$_3$O$^+$-O$_2$]
photoexcitation (Cl$^-$ from BB extracts), H$_2$O$_2$ oxidation, BC-catalyzed oxidation, reactive
nitrogen species oxidation, and organics-driven pathways e.g., HCHO, photosensitizing
components, organic peroxide, and TMI-organic oxidation (Ye et al., 2023).
Since there was no nitrate peak in our Raman spectra in all experiments, the potential impact
from nitrate photolysis was excluded. Besides, the significantly low Cl$^-$ concentration (0.0002-
0.001M) in the original BB extracts (compared to 1M NaCl, Table S1) has minimized the
influence of chloride photoexcitation of [Cl$^-$-H$_3$O$^+$-O$_2$] (Cl$^-$ from BB extracts) on the sulfate
formation. Reactive nitrogen species e.g., NO$_x$, HONO and NH$_3$ were neither introduced nor
detected in our system, indicating that the oxidation pathway involving reactive nitrogen
species as insignificant. Additionally, the water extraction process has excluded the possibility
of BC-catalyzed oxidation. The absence of sulfate formation in dark conditions ruled out the
involvement of direct H$_2$O$_2$ oxidation and organic peroxide oxidation pathways. The
concentrations of TMI did not exhibit a consistent relationship with the sulfate formation
observed in both BB$_F$-NaCl and BB$_A$-NaCl droplets (Figure S2), suggesting that the TMI-
catalyzed oxidation pathway may not be responsible for the observed phenomenon. Therefore,
the most probable reason for the enhancement of sulfate formation by BB-NaCl droplets over
NaCl droplets would be the photosensitizing components. State-of-the-art mass spectrometry





analysis including UHPLC-Orbitrap-MS and GC×GC-MS showed the existence of possible
photosensitizers such as PAHs (e.g., fluoranthene, pyrene, cyclopenta[cd]pyrene, 4-
methylphenanthrene, benzo[a]pyrene, perylene, Table S2) and aromatic carbonyls (SyrAld, VL,
3,4-dimethoxybenzaldehyde, acetophenone, acetosyringone, Table S2). Photosensitizing
components can directly or indirectly (by forming secondary oxidants in the presence of oxygen)
oxidize S(IV) to S(VI). Wang et al. (2020b) proposed a direct oxidation process of S(IV) to
sulfate by excited triplet states of photosensitizers (PS*). To explore the contribution of the
direct PS* oxidation on sulfate formation, we performed the same sets of experiments in $N_2$-
saturated condition, shown in Figure S3. The BB-NaCl droplets showed only direct PS*
oxidation contribution of 3.6% to 22.7%, highlighting the predominant role of secondary
oxidants (Tang et al., 2023). For $BB_F$-NaCl droplets, the contribution of direct PS* followed
the trend of $WS_F$-NaCl (22.7%) > $RS_F$-NaCl (15.7%) > $CS_F$-NaCl (7.0%), while for $BB_A$-NaCl
droplets, $WS_A$-NaCl (10.2%) > $CS_A$-NaCl (6.7%) > $RS_A$-NaCl (3.6%) was observed. In
summary, regardless of whether fresh or aged, the secondary oxidants triggered by indirect PS*
oxidation were the main reason for sulfate formation, highlighting the importance of $O_2$ in PS*
mediated oxidation processes.

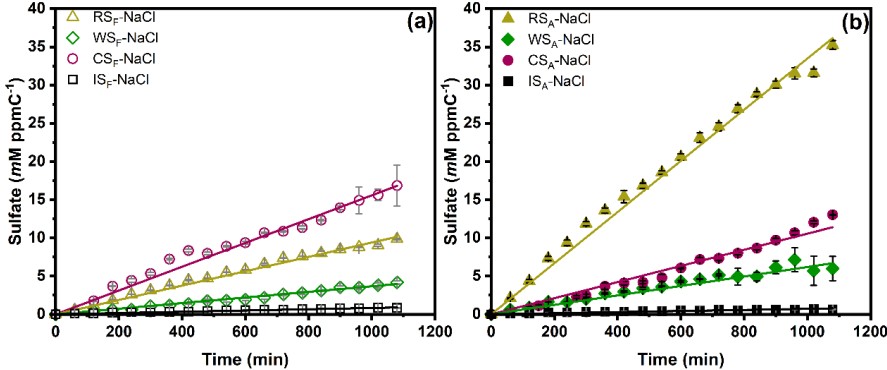


Figure 1. Sulfate production under different droplet compositions as a function of time: (a) fresh
BB-NaCl droplets; (b) aged BB-NaCl droplets in air at 80% RH. RS, WS, CS and IS represent
rice straw, wheat straw, corn straw and incense burning, respectively. The subscripts F and A
represent fresh and aged, respectively.
Table 1. Sulfate formation rate constant ($k_{so_4^{2-}}$), reactive ($\gamma_{SO_2}$) and normalized $SO_2$ uptake
coefficient ($n\gamma_{SO_2}$) of various particle compositions at 80% RH. Sulfate formation rate ($k_{so_4^{2-}}$)
for aqueous phase reactions using different BB extracts and model compounds. 1, 10, 100 and
200 represent the concentration of different compounds (in ppm).

| Particle Composition | $k_{so_4^{2-}}$ | $\gamma_{SO_2}$ | $n\gamma_{SO_2}$ [a] |
|---|---|---|---|
| | ($\mu$M min$^{-1}$ ppmC$^{-1}$) | | ppmC$^{-1}$ |
| $RS_F$-NaCl | $9.4 \pm 0.10$ | $(2.2 \pm 0.023) \times 10^{-6}$ | $(6.2 \pm 0.066) \times 10^{-8}$ |
| $WS_F$-NaCl | $3.7 \pm 0.048$ | $(0.66 \pm 0.0086) \times 10^{-6}$ | $(2.0 \pm 0.027) \times 10^{-8}$ |



| | | | |
|---|---|---|---|
| $CS_F$-NaCl | $15.6 \pm 0.11$ | $(2.0 \pm 0.015) \times 10^{-6}$ | $(8.8 \pm 0.065) \times 10^{-8}$ |
| $IS_F$-NaCl | $0.83 \pm 0.011$ | $(1.7 \pm 0.034) \times 10^{-6}$ | $(0.61 \pm 0.012) \times 10^{-8}$ |
| $RS_A$-NaCl | $33.5 \pm 0.38$ | $(6.6 \pm 0.074) \times 10^{-6}$ | $(21.5 \pm 0.24) \times 10^{-8}$ |
| $WS_A$-NaCl | $6.2 \pm 0.18$ | $(0.92 \pm 0.027) \times 10^{-6}$ | $(3.5 \pm 0.10) \times 10^{-8}$ |
| $CS_A$-NaCl | $10.6 \pm 0.23$ | $(1.0 \pm 0.023) \times 10^{-6}$ | $(6.2 \pm 0.13) \times 10^{-8}$ |
| $IS_A$-NaCl | $0.72 \pm 0.026$ | $(1.3 \pm 0.052) \times 10^{-6}$ | $(0.46 \pm 0.017) \times 10^{-8}$ |
| Aqueous Reactions | Concentration | $k_{so_4^{2-}}$ | $k_{so_4^{2-}}$ |
| | (ppm) | (ppm min$^{-1}$) | ($\mu$M min$^{-1}$) |
| $RS_F$ | 1 | 0.31 | 3.2 |
| $RS_F$-NaCl | 1-100 | 0.16 | 1.6 |
| $RS_F$-NaCl | 1-200 | 0.085 | 0.9 |
| $WS_F$ | 1 | 0.19 | 2.0 |
| $CS_F$ | 1 | 0.25 | 2.6 |
| $IS_F$ | 1 | 0.19 | 2.0 |
| $RS_A$ | 1 | 0.33 | 3.4 |
| $RS_A$-NaCl | 1-100 | 0.37 | 3.8 |
| $RS_A$-NaCl | 1-200 | 0.63 | 6.4 |
| $WS_A$ | 1 | 0.26 | 2.7 |
| $CS_A$ | 1 | 0.33 | 3.4 |
| $IS_A$ | 1 | 0.080 | 0.82 |
| NaCl | 100 | 0.051 | 0.52 |
| NaCl | 200 | 0.079 | 0.81 |
| SyrAld | 1 | 0.15 | 1.5 |
| SyrAld-Pyz | 1-1 | 0.68 | 7.1 |
| SyrAld-Pyz-NaCl | 1-1-10 | 0.67 | 6.9 |
| SyrAld-Pyz-NaCl | 1-1-100 | 0.55 | 5.7 |
| SyrAld-Pyz-NaCl | 1-1-200 | 0.50 | 5.2 |
| SyrAld-4-NC | 1-1 | 0.11 | 1.1 |
| SyrAld-4-NC-NaCl | 1-1-10 | 0.13 | 1.4 |
| SyrAld-4-NC- | 1-1-100 | 0.13 | 1.4 |



| | | | |
|---|---|---|---|
| NaCl | | | |
| SyrAld-4-NC-NaCl | 1-1-200 | 0.15 | 1.5 |
| SyrAld-NaCl | 1-10 | 0.11 | 1.1 |
| SyrAld-NaCl | 1-100 | 0.17 | 1.8 |
| SyrAld-NaCl | 1-200 | 0.17 | 1.7 |
| VL | 1 | 0.26 | 2.7 |
| VL-Pyz | 1-10 | 0.61 | 6.4 |
| VL-Pyz-NaCl | 1-1-10 | 0.55 | 5.8 |
| VL-Pyz-NaCl | 1-1-100 | 0.43 | 4.5 |
| VL-Pyz-NaCl | 1-1-200 | 0.42 | 4.3 |
| VL-4-NC | 1-1 | 0.17 | 1.7 |
| VL-4-NC-NaCl | 1-1-10 | 0.22 | 2.3 |
| VL-4-NC-NaCl | 1-1-100 | 0.27 | 2.7 |
| VL-4-NC-NaCl | 1-1-200 | 0.23 | 2.4 |
| VL-NaCl | 1-10 | 0.25 | 2.6 |
| VL-NaCl | 1-100 | 0.26 | 2.7 |
| VL-NaCl | 1-200 | 0.28 | 2.9 |

[a]The $\gamma_{SO_2}$ was normalized by the initial TOC concentration (ppmC), i.e., $n\gamma_{SO_2} = \gamma_{SO_2}/TOC$


**3.2 Aging effects on sulfate formation across various BB materials**


To investigate the aging effects across various BB materials, we aged the collected BB filters
by irradiating with UV lights (185 nm and 254 nm) (Tang et al., 2023). Figure S4 exhibits the
differences in sulfate formation rates of different fresh and aged BB materials. RS and WS show
sulfate formation enhancement, while CS and IS show reduction after aging. Figure 2(a) shows
that the 18h sulfate enhancement factor (Sulfate$_A$/Sulfate$_F$) followed the trend of RS-NaCl
(3.6) >WS-NaCl (1.4) > CS-NaCl (0.8) ≈ IS-NaCl (0.8), which is neither consistent with the
trends of sulfate formation for BB$_F$-NaCl nor BB$_A$-NaCl, indicating that aging processes have
different influence on sulfate formation towards BB materials. A similar trend was found for
$n\gamma_{SO_2}$, showing the highest and lowest sulfate enhancement for RS-NaCl (3.5) and IS-NaCl
(0.7), respectively.





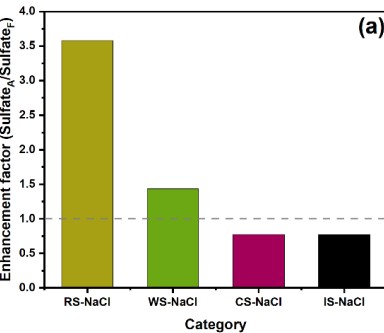
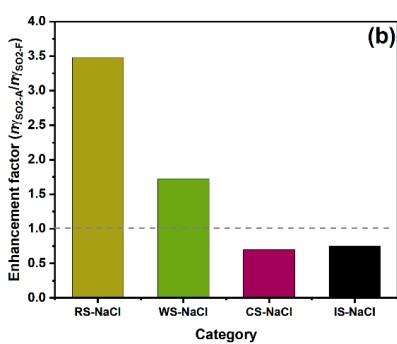

Figure 2. Enhancement factor of (a) sulfate and (b) normalized $SO_2$ uptake coefficient $n\gamma_{SO2}$ between fresh and aged BB-NaCl droplets.

We also performed aqueous reactions using fresh/aged BB extracts to investigate the aging effects on the sulfate formation (Figure S7). The sulfate formation rate ($k_{so_4^{2-}}$) for different BB extracts during initial photoinduced experiments ranged from 0.8 to 3.4 µM min$^{-1}$. The $k_{so_4^{2-}}$ obtained in bulk-phase reactions were a magnitude lower than that of the droplets (Table 1), which is consistent with previous studies (Wang et al., 2024b; Zhang and Chan, 2024). Wang et al. (2024b) discovered sulfate formation rate magnitudes higher at air-water interface (AWI) than conventional bulk-phase reactions. They attributed this to accelerated electron transfer process at AWI, where $^3$PS* ($^3$HULIS* in their case) can accept electrons from $HSO_3^-$ in a more efficient way due to their incomplete solvent cages. Zhang and Chan (2024) fitted a ~3 orders of magnitude higher rate constant of IC*+Cl$^-$ (~10$^8$ M$^{-1}$s$^{-1}$) in particle phase than bulk-phase rate constant (~10$^5$ M$^{-1}$s$^{-1}$) (Gemayel et al., 2021; Woods et al., 2020). They attributed the enhanced sulfate formation to the expedited reactions between $^3$PS* and chloride ions to form reactive chlorine species, facilitated by the decreased solvation of chloride and $^3$PS* at the AWI (Zhang and Chan, 2024). Many studies have demonstrated that chloride ions, bisulfite ions and surfactant-like PS have the propensity to reside at the AWI of droplets, primarily driven by polarization interactions. This promotes enlarged bond dipole moments and ordered alignment of reactant molecules, resulting in reduced entropy and heightened free energy of the initial state (Jungwirth and Tobias, 2002, 2006; Ruiz-Lopez et al., 2020; Tinel et al., 2016; Yan et al., 2016; Fu et al., 2015). Other factors, e.g., S(IV) concentration (8 ppm gaseous $SO_2$ in droplet experiment and 100 ppm $HSO_3^-$ in aqueous reactions) and the addition of NaCl (1M NaCl addition in droplets and no NaCl addition in aqueous reactions) may also contribute to the high sulfate formation rate in droplet experiments. In this study, the more efficient sulfate formation in droplet experiments than bulk solutions can potentially be attributed to the accelerated reactions induced by photosensitizers at the AWI, intensity variance in droplets and aqueous solution, concentrations difference in S(IV) and the addition of NaCl. However, the detailed mechanisms of the accelerated sulfate formation in droplets than bulk are still uncertain and out of the scope of this paper, and more research should be performed in the future.

In bulk experiments, all BB extracts have higher $k_{so_4^{2-}}$ after aging. The increased sulfate formation of BB extracts after aging may be due to changes in their chemical compositions.





Compared to $RS_F$ (28.3% for CHON- and 67.3% for CHN+ in total intensity), $RS_A$ has higher
CHON- (36.1%) and CHN+ (88.3%) percentages. Zhao et al. (2022) observed a slight increase
in CHON percentage for RS from 53.4% to 56.2% after aging. Similar trend was observed for
CS extracts, where CHON- and CHN+ percentage increases from 26.7% and 65.2% to 31.5%
and 68.8%, respectively, after aging. As chromophoric compounds are present in brown carbon
(BrC) (Laskin et al., 2015), we constrained the DBE values ($0.5c \leq DBE \leq 0.9c$) to semi-
qualitatively distinguish chromophores in the dissolved BrC (Lin et al., 2018). Higher amounts
of CHON- species were found in $RS_{A-BrC}$ (41.9%) and $CS_{A-BrC}$ (35.5%) than $RS_{F-BrC}$ (32.3%)
and $CS_{F-BrC}$ (34.7%). One of the key categories of CHON- is nitrated aromatics, which have
been widely identified in lab-generated BB smoke (Huang et al., 2022b; Wang et al., 2017a;
Zhang et al., 2022; Xie et al., 2019) and field campaigns (Salvador et al., 2020; Mohr et al.,
2013; Chen et al., 2022). A series of CHON- species, e.g., $C_6H_5NO_3$, $C_6H_5NO_4$, $C_7H_7NO_3$, and
$C_8H_9NO_3$, which were tentatively identified as nitrophenol, nitrocatechol, methyl-nitrophenol,
and dimethyl-nitrophenol, have been detected in our BB extracts. Nitrophenols photolysis has
been found to be a potential source of OH radicals (Sangwan and Zhu, 2018; Guo and Li, 2023;
Cheng et al., 2009; Sangwan and Zhu, 2016). Therefore, the increase in sulfate formation by
$RS_A$ and $CS_A$ may partially be related to the more oxidants generated by nitrophenol photolysis.
Approximately 80% of the CHN+ species identified exhibited a diatomic nitrogen composition
in their molecular formula. The precise determination of the molecular structures of these
compounds solely based on elemental composition is challenging due to the presence of stable
isomers. However, the N-bases, which contain two nitrogen atoms, can be attributed to various
N-heterocyclic alkaloids. For example, homologs of $C_5H_6N_2(CH_2)_n$ were likely pyrazine,
pyrimidine or amino pyridine, which were composed of six-membered heterocyclic rings with
N atoms and alkyl side chains (Lin et al., 2012; Laskin et al., 2009). $C_5H_8N_2(CH_2)_n$ were likely
alkyl-substituted imidazole compounds, featuring a five-membered heterocyclic ring with two
nitrogen atoms as the core structure and alkyl side chains (Lin et al., 2012; Laskin et al., 2009).
For $C_7H_6N_2(CH_2)_n$ homologs, the core skeleton was $C_7H_6N_2$, with an $AI_{mod}$ of 0.8, indicating its
distinctive characteristics of compounds containing fused five-membered and six-membered
rings, such as benzimidazole or indazole (Wang et al., 2017b). Redox-inactive heterocyclic
nitrogen-containing bases, e.g., pyridine, imidazole, and their derivatives, have been shown to
enhance the redox activity of humic-like substances (HULIS) fraction by hydrogen-atom
transfer, with the degree of enhancement directly correlated to their concentration (Dou et al.,
2015; Kipp et al., 2004). Thus, the increased CHN+ percentage may also contribute to the
enhanced sulfate formation of $RS_A$ and $CS_A$ by acting as a H-bond acceptor to facilitate the
$^3PS^*$-mediated oxidation by generating more oxidants.
However, the CHON- and CHN+ percentages in $WS_A$ were lower than $WS_F$, indicating that the
sulfate enhancement in $WS_A$ was not due to the CHON and CHN species. Instead, CHO-
accounted for higher proportion in $WS_A$ (68.5%) and $WS_{A-BrC}$ (68.9%) than $WS_F$ (65.0%) and
$WS_{F-BrC}$ (64.8%).This aligns with a prior AMS study, showing increased CHO proportions in
aged wheat burning emissions (Fang et al., 2017). We suppose that CHO- compounds,
particularly photosensitizing compounds with carbonyl groups, would explain the difference of
sulfate formation in WS extracts (Gómez Alvarez et al., 2012; Mabato et al., 2023; Felber et al.,
2020; Fu et al., 2015). Therefore, we filtered the chemical formula of CHO- species from



UHPLC-Orbitrap-HRMS by applying maximum carbonyl ratio (MCR) (Zhang et al., 2021),
H/C, O/C as well as modified aromaticity index ($AI_{mod}$) to focus on potential photosensitizers
(Zherebker et al., 2022; Koch and Dittmar, 2006). In short, molecular formula were classified
into six groups, namely, condensed aromatics ($AI_{mod} \geq 0.67$), polyphenolics ($0.50 < AI_{mod} < 0.67$),
highly unsaturated and phenolic compounds ($AI_{mod} \leq 0.5$, H/C<1.5), aliphatics (H/C≥1.5,
O/C≤0.9, N=0), peptide-like compounds (H/C≥1.5, O/C≤0.9, N>0) and sugar-like compounds
(H/C≥1.5, O/C>0.9), details can be found in Text S1. As aliphatics, peptide-like compounds
and sugar-like compounds are unlikely to be photosensitizers, we exclude them as potential PS.
By applying a data filtration process involving CHO-, condensed aromatics, polyphenolics,
highly unsaturated and phenolic compounds based on the criteria abovementioned, and
MCR≥0.9, 52.6% and 49.7% of the compounds (by intensity) were selected by $WS_A$ and $WS_F$,
respectively. The main compositional difference lies in polyphenolics, comprising 26.3% and
21.8% of $WS_A$ and $WS_F$ respectively. Therefore, the higher sulfate formation in $WS_A$ may be
related to the higher contributions of the polyphenolics, e.g., $C_8H_8O_3$.
**3.3 Effects of Chloride and Nitrogen-containing Species on Sulfate Formation**
Unlike the droplet experiments where RS-NaCl has the highest sulfate enhancement factor after
aging, aqueous reaction results (without NaCl) show a sulfate enhancement trend of
WS>CS>RS>IS, suggesting that chloride may take effect in the droplet experiments, especially
in RS-NaCl system. Therefore, bulk reaction experiments using rice straw (RS) extracts as an
example were performed with 100-200 ppm NaCl additions, where the NaCl to TOC ratio
ranged from 100:1 to 200:1 to match the 100:1 to 1000:1 range in droplet experiments, in order
to evaluate the effects of chloride on sulfate formation. Interestingly, incorporating NaCl
yielded contrasting results for $RS_F$ and $RS_A$ (Figure 3). While the addition of NaCl enhanced
sulfate formation in $RS_A$, it showed the opposite trend in $RS_F$. The nature of the cations and
ionic strength may affect the sulfate formation rate; however, previous studies have indicated
that their effects are negligible (Zhang and Chan, 2024; Parker and Mitch, 2016). The opposite
effect of the NaCl addition on $RS_F$ and $RS_A$, to some extent, explain the significantly higher
sulfate and $SO_2$ uptake coefficient enhancement factor for RS-NaCl in Fig. 2. Compared to the
RS-based system, NaCl control experiment showed minimum (but non-zero) sulfate formation
(Table 1 and Figure 3). On one hand, it confirmed that chloride participated in the sulfate
formation in bulk reactions, possibly by forming Cl and OH radicals in the presence of air and
water (Cao et al., 2024; Tang et al., 2023; Zhang and Chan, 2024). On the other hand, the
opposite trend of $Cl^-$ effects on $RS_F$ and $RS_A$ reflects its complex interactions with BB extracts
under light and air. While direct reaction between S(IV) species and triplet states of
photosensitizers (PS*) may occur (Wang et al., 2020b), other pathways, i.e., interactions among
halide ions, photosensitizers and oxygen should also be considered. PS in BB extracts can
absorb solar radiation and form triplet-state photosensitizer ($^3PS^*$), which can then react with
molecular oxygen and form singlet-state oxygen $^1O_2^*$ through energy transfer. $^3PS^*$ can also
react with H-donor, typically organic acids (RH, e.g., vanillic acid, succinic acid, azelaic acid,
glutaric acid, sorbic acid, salicylic acid, Table S3) through H transfer reactions, and form a ketyl
radical (PSH•) and an alkyl or phenoxy radical (R•). PSH• and R• can then participate in a series
of reactions to form OH•, $HO_2$•, $H_2O_2$ and $O_2$•⁻. In the presence of a large excess of $Cl^-$, $Cl^-$ can
act as an electron donor, and react with $^3PS^*$, forming a Cl• and a deprotonated ketyl radical





(PS•) (Jammoul et al., 2009). Further reactions are similar to the abovementioned reactions,
including the formation of reactive chlorine species (RCS, i.e., Cl•, $Cl_2^{•-}$ and ClOH•⁻) and
reactive oxygen species (ROS, i.e., OH•, $HO_2$•, $H_2O_2$ and $O_2^{•-}$). These RCS and ROS
simultaneously contribute to S(IV) oxidation to S(VI) (Zhang and Chan, 2024).
Statistical analysis using the Pearson correlation coefficient revealed that the concentrations of
CHO, CHON, and CHN species exhibited significant correlations ($|R|>0.5$) with the sulfate
formation rate ($p < 0.05$, Figure S9). As PS can be the main CHO species contributing to sulfate
formation, N-containing organic compounds (NOCs), i.e., CHN and CHON species, may affect
the chloride contribution on sulfate formation rate. Therefore, we selected SyrAld and VL as
model CHO (PS), pyrazine (Pyz) as a model CHN, and 4-nitrocatechol (4-NC) as a model
CHON to elucidate how potential chemical compounds can alter the effects of chloride on
sulfate formation rate by studying the CHO+Cl⁻, CHO+CHN+Cl⁻, and CHO+CHON+Cl⁻
systems. For SyrAld and VL, as the $[Cl^-]_0/[PS]_0$ increases, $k_{so_4^{2-}}$ initially decreases and then
increases. The initial decrease of $k_{so_4^{2-}}$ may be attributed to the quenching of $^3$PS* by electron
transfer from Cl⁻ or loss of OH radials by forming ClOH•⁻ through reaction of OH•+Cl⁻↔ClOH•⁻
(Anastasio and Newberg, 2007). Excessive chloride (e.g. 100 and 200 ppm) may generate Cl
and OH radicals through photoexcitation in the presence of air and water and compensate for
the loss of $^3$PS* or OH radicals. Previous studies have shown controversial influence of halides
on the photosensitized oxidation of organic compounds or bisulfite. Parker and Mitch (2016)
and Zhang et al. (2023) attributed the significantly higher photodegradation of dienes,
thioethers and acetaminophen to the formation of reactive halogen species generated by the
reactions of PS and halides. Zhang and Chan (2024) reported that $[Cl^-/PS]_0$ in the range of 1:2
to 4:1 did not lead to significant difference in sulfate formation, possibly due to the insufficient
Cl⁻ concentration in triggering the interplay between PS and Cl⁻. The differences between the
current results and the aforementioned study might be attributed to the higher $[Cl^-/PS]_0$ (up to
1:200) which may have been sufficient to initiate the relevant reactions, as well as the difference
in photosensitizing capacities of the PS studied (triplet quantum yield of $0.86 \pm 0.05$ for 2-IC
and $0.21 \pm 0.01$ for VL) (Felber et al., 2021; 2020). Safiarian et al. (2023) reported that
increasing chloride concentrations facilitated anthracene photosensitization by producing high-
level reactive oxygen species (ROS). Wang et al. (2023a) found that the effects of chloride on
sulfate formation depended on the specific PS: enhancing sulfate production for benzophenone
(BP) and 3,4-dimethoxybenzaldehyde (DMB), but decreasing it for 1,4-naphthoquinone.
When incorporating CHN species, a 2-3-fold $k_{so_4^{2-}}$ was observed, due to the enhanced H
transfer by CHN acting as H-bond acceptor (Dou et al., 2015). With the addition of NaCl, the
enhanced H-transfer effect by CHN was inhibited, possibly due to the consumption of $^3$PS* by
Cl⁻. The addition of model CHON species into PS decreased $k_{so_4^{2-}}$, due to the consumption of
$^3$PS* by CHON species, in agreement with Wang et al. (2023b) who reported increased effective
quantum yield of 4-NC when co-photolysis with VL. Further addition of NaCl increased the
$k_{so_4^{2-}}$, possibly due to the consumption of 4-NC by RCS (Wang et al., 2024a), which, to some
extent, reduced the loss of $^3$PS*. Generally, the addition of chloride increased $k_{so_4^{2-}}$ of PS-
CHON but decreased $k_{so_4^{2-}}$ of PS-CHN. However, the ambient air is characterized by the
presence of tens of thousands of chemical compounds. As a result, the interplay among this
diverse array of species may occur in ways that exceed current understanding, necessitating



additional research to investigate the interactions between different organic compounds more
thoroughly.

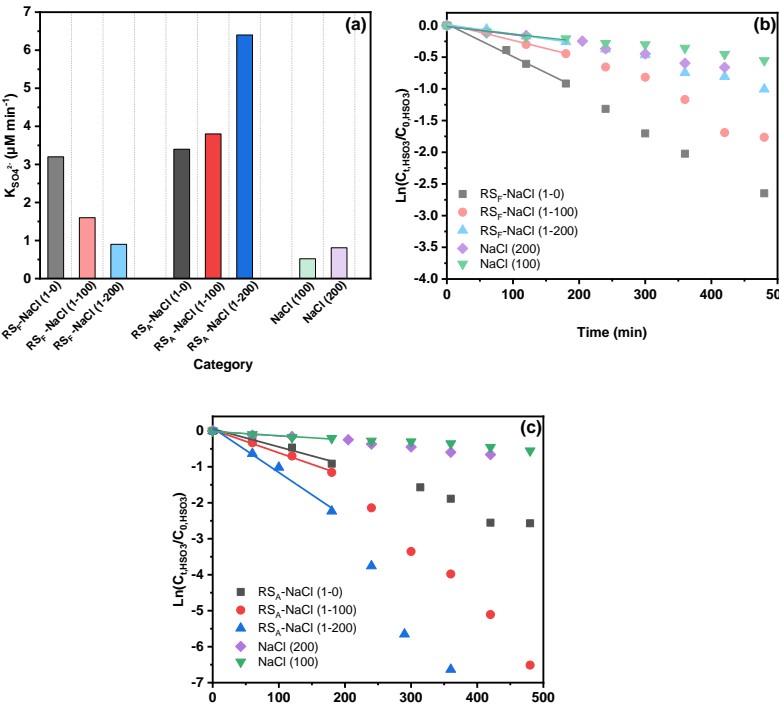


Figure 3. (a) Sulfate formation rate and (b) (c) bisulfite decay in RS-NaCl system. 1-0, 1-100, and
1-200 refer to the concentration ratios of $TOC_{RS}$ and NaCl, in which 1, 100, 200 represent 1 ppm,
100 ppm and 200 ppm, respectively.

## 3.4 Proposed mechanism for sulfate formation

A conceptual diagram of PS and chloride mediated ROS and RCS production in the oxidation
of S (IV) to S (VI) was shown in Fig. 4. Initially, the photosensitizers (PS, e.g., SyrAld and VL)
absorb solar radiation and produce the singlet state $^1$PS*, which then undergo a spin conversion
through intersystem crossing, leading to the formation of the triplet state $^3$PS*. The $^3$PS* can
react with molecular oxygen through energy transfer and generate singlet state$^1$O$_2$*, while the
$^3$PS* returns to ground state. The $^1$O$_2$* can then transform to O$_2$$^•$ via electron transfer. The
$^3$PS*can also react with an H donor (RH, e.g., organic acids, syringol, guaiacol, Table S3),
leading to the formation of alkyl or phenoxy radical (R$^•$) and a ketyl radical (PSH$^•$). R$^•$ can react
with O$_2$ and form RO$_2$ radicals while PSH$^•$ can transfer an H atom to O$_2$ and form HO$_2$$^•$,
returning to its ground state PS. Additionally, $^3$PS* can react with an electron donor, e.g., Cl$^-$,



and form chlorine radicals and PS$^{\bullet-}$. The formed PS$^{\bullet-}$ then reacts with $O_2$ and form $O_2^{\bullet-}$, which
undergoes a series of reactions and form $HO_2^{\bullet}$, $H_2O_2$ and OH$^{\bullet}$. The above-mentioned reactions
are the main processes in ROS pathway. Recently, Zhang and Chan(2024) have proposed that
the reactive chlorine species (RCS) would contribute to sulfate formation. Cao et al. (2024)
proposed a mechanism of OH and Cl radicals formation by [Cl$^-$-$H_3O^+$-$O_2$] under light irradiation
through an electron transfer process. Our results also demonstrate that the addition of Cl$^-$ will
affect the oxidation process of S(VI) (Figures 3, S10-S12). Combining the above, the RCS
pathway was shown in yellow arrows in Figure 4. The Cl$^{\bullet}$ can be formed in two pathways,
photoexcitation of the [Cl$^-$-$H_3O^+$-$O_2$] complex that generates Cl radicals in deliquescent BB-
NaCl droplets or aqueous BB-NaCl solution (Cao et al., 2024), and PS* mediated Cl$^{\bullet}$ formation
via electron transfer by Cl$^-$ (Corral Arroyo et al., 2019). The formed Cl$^{\bullet}$ can then react with each
other through radical-radical reactions and produce molecular $Cl_2$. The Cl$^{\bullet}$ can also react with
Cl$^-$ or $Cl_2^{\bullet-}$, forming $Cl_2^{\bullet-}$ or $Cl_2$. Cl$^{\bullet}$ and $Cl_2^{\bullet-}$ can also react with OH and form HOCl. $^3$PS*
itself can also oxidize the S(IV) (e.g., dissolved $SO_2$ or bisulfite) to S(VI). However,
significantly lower sulfate formation was found in the presence of $N_2$ compared to air condition
(Figure S3), highlighting the importance of secondary oxidants compared to direct PS*
oxidation. As a consequence, these reactive species, e.g., OH$^{\bullet}$/$HO_2^{\bullet}$/$O_2^{\bullet-}$ and Cl$^{\bullet}$/$Cl_2^{\bullet-}$ may all
participate in the oxidation of S(IV) to S(VI). In addition, the nitrogen-containing heterocyclic
compounds such as pyrazine can act as H-bonding acceptor and facilitate the H transfer, which
then generates more ROS (Dou et al., 2015). In light of the absence of substantial fluctuations
in chloride concentration (Figure S13 and S14, insignificant chloride concentration change was
found even in 10 ppm NaCl addition), it is postulated that chloride ions may function as a
reactive medium rather than as direct reactants. In this proposed scenario, the Cl radicals and
$Cl_2^{\bullet-}$ intermediates generated during the reaction subsequently undergo reversion back to Cl$^-$
ions, thereby maintaining a relatively constant Cl$^-$ concentration throughout the experimental
observations. Note that although ROS and RCS pathways both contribute to the oxidation from
S(IV) to S(IV), they may act as competitive relationships due to the co-consumption of PS*.
Therefore, different Cl effects may occur regarding various combinations of reactants (Figure
3, promoting effect in RS$_A$, inhibiting effects on RS$_F$).





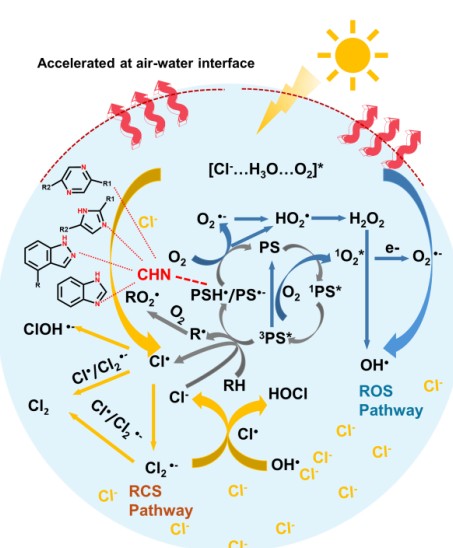


Figure 4. Conceptual diagram of PS and chloride mediated ROS and RCS production, which
participates in the oxidation processes from S(IV) to S(VI)
**4 Atmospheric Implication**
This study provided laboratory evidence that the photosensitizers in biomass burning extracts
can enhance the sulfate formation in NaCl particles, primarily by triggering the formation of
secondary oxidants under light and air, with less contribution of direct photosensitization via
triplets (evidenced by $N_2$ atmosphere, Figure S3). The sulfate formation rate of $BB_F$-NaCl
particles were ~10 folds higher that of $IS_F$-NaCl, following the trends of $CS_F$-NaCl>$RS_F$-
NaCl>$WS_F$-NaCl>$IS_F$-NaCl. Upon UV exposure, the sulfate formation trends shifted to $RS_A$-
NaCl>$CS_A$-NaCl>$WS_A$-NaCl>$IS_A$-NaCl, which might be explained by the effects of chloride
(evidenced by aqueous reactions, Figure 3 and Table 1). Interestingly, the incorporation of $Cl^-$
into bulk solutions increased the sulfate formation rate in $RS_A$, while decreased it in $RS_F$. This
seems to be different from our group's previous work where no significant sulfate formation
rate was found with the addition of $Cl^-$ (Zhang and Chan, 2024). The difference can be
explained by the following reasons: 1) differences in PS/$Cl^-$, the prior study might use an
insufficient PS/$Cl^-$ ratio (2:1-1:4) while the current one significantly expands it to 1:200. 2)
differences in photosensitizing capacity: the former study used a strong PS, while the current
study focused on the real BB (using TOC as metric, with only a small portion of TOC
considered as PS). 3) the complexity of the reaction system, the former study focused on mixing
two individual species, while in real BB extracts, more complicated reactions may occur.
Furthermore, our results using model PS show that although additional model CHN species
would increase the sulfate formation by expedited H transfer via acting as H-bond acceptor, the
addition of chloride could inhibit the sulfate formation rate, suggesting that the RCS pathway
was less efficient in sulfate formation compared to ROS pathway in PS-CHN bulk system
(Figure S10 and S11).



Previous studies have detected a significant proportion of NOCs, including nitroaromatics (CHON) and reduced nitrogen species (CHN) in biomass burning plumes, wildfires and ambient samples (Zhong et al., 2024; Wang et al., 2017b; Song et al., 2022; Cai et al., 2020). These NOCs are considered as ubiquitous contributor to BrC, and can affect global climate and human health. Moreover, recent research has discovered aerosol pollution in marine background regions, with high levels of NOCs when air masses are transported from wildfires or biomass burning events in nearby (Zhong et al., 2024; Qin et al., 2024). These NOCs, combined with reactive gases, may mix with sea-salt aerosols and impact regional air quality in coastal zones. While our prior study has examined the potential interplay between chloride and PS at limited mixing ratios (up to 4:1 in bulk solution) (Zhang and Chan, 2024), this work expanded the Cl⁻/PS ratio to a broader range (200:1) and systematically identified the interactions among different organics, including PS, NOCs, and chloride, using sulfate formation as a compass. This highlights the importance to study secondary aerosol formation in mixed experimental system under air pollution complex. Our work suggests that in coastal regions heavily influenced by anthropogenic emissions like biomass burning, especially those near the rice-growing regions or affected by transported wildfire smoke, such as Guangdong, Fujian and Taiwan, the transported BB plumes together with the high RH (Cheung et al., 2015) and abundant reactive gases, would play an inevitable role in sulfate and potentially secondary organic aerosol formation.

**Data availability**

Datasets are available upon request to the corresponding author, Chak K. Chan (chak.chan@kaust.edu.sa).

**Author contributions**

RT and CC conceptualized and designed the study. YQ and YC collected the samples. RT performed the experiments, data analysis and wrote the draft. JM provided assistance in data processing. All the authors reviewed, edited and contributed to the scientific discussions.

**Competing interests**

The authors declare no conflicts of interest.

**Acknowledgments**

We gratefully acknowledge the support from the Hong Kong Research Grants Council (No. 11314222), the National Natural Science Foundation of China (42107115), and the Natural Science Foundation of Shandong Province, China (ZR2021QD111). The authors also thank the University Research Facility in Chemical and Environmental Analysis (UCEA) at The Hong Kong Polytechnic University for the use of its UHPLC-HESI-Orbitrap Mass Spectrometer and Dr Sirius Tse and Dr Chi Hang Chow for assistance with sample analyses.

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

Index with Improved Estimation of Carboxyl Group Contribution for Biogeochemical Studies,
Environmental Science & Technology, 56, 2729-2737, 10.1021/acs.est.1c04575, 2022.
Zhong, S., Liu, R., Yue, S., Wang, P., Zhang, Q., Ma, C., Deng, J., Qi, Y., Zhu, J., and Liu, C.-Q.: Peatland
Wildfires Enhance Nitrogen-Containing Organic Compounds in Marine Aerosols over the Western
Pacific, Environmental Science & Technology, 2024.
Zhou, L., Liang, Z., Mabato, B. R. G., Cuevas, R. A. I., Tang, R., Li, M., Cheng, C., and Chan, C. K.:
Sulfate formation via aerosol-phase SO2 oxidation by model biomass burning photosensitizers: 3,4-
dimethoxybenzaldehyde, vanillin and syringaldehyde using single-particle mixing-state analysis, Atmos.
Chem. Phys., 23, 5251-5261, 10.5194/acp-23-5251-2023, 2023.