# Peer review of "Enhanced Sulfate Formation in Mixed Biomass Burning and Sea-salt"

_EGUsphere, 2024_

## Author Comment (AC1)

**Point-by-point response to reviewers**

**RC1:**

*Tang et al. reported the enhancement of sulfate production from photosensitization in biomass burning-NaCl droplets. The effects of atmospheric aging and chloride ions in photosensitization-initiated sulfate formation and the corresponding influencing mechanism were also discussed. The experimental results are solid and reliable, but the discussion is not deep enough. I have outlined some of the shortcomings below.*

*In this manuscript, sulfate formation in the atmosphere was investigated by mixing biomass burning and sea-salt particles. My major concern is about the environmental relevant of the designed experiments. The authors showed in detail the reported formation mechanism of sulfates and the prevalence of biomass burning in the Introduction. It is too tedious. The author does not logically explain the importance and necessity of studying sulfate formation by mixing real biomass burning extracts and NaCl. In addition, the relationship between photosensitization and biomass burning should be shown in a few words.*

**Response:** We appreciate the reviewer's thoughtful feedback on our manuscript.

We have shortened our introduction and incorporated the importance and necessity of studying sulfate formation by mixing real BB extracts and NaCl. We also showed the link between photosensitization and biomass burning in the introduction. Overall, when BB air mass mixes with marine air mass, the interactions of BB and sea-salt components are important to atmospheric chemistry, especially multiphase reactions due to the high humidity. The introduction has been modified to the following (lines 36-89):

[revised manuscript text omitted]

The manuscript has been altered accordingly.

Minor comments:

*1.        Please check carefully English statement in the whole manuscript.*

**Response:**

We have conducted a comprehensive check of the manuscript and made relevant modifications to enhance its quality.

*2.        Line 98–100. As mentioned by the authors, the aim of this study includes three points, but the title only covered one point.*

**Response:**

We have modified the title to "Enhanced Sulfate Formation in Mixed Biomass Burning and Sea-salt Interactions Mediated by Photosensitization: Effects of Chloride, Nitrogen-containing Compounds and Atmospheric Aging" to cover all the aspects raised in the abstract.

*3.        Line 100. "investigating".*

**Response:**

We have changed the "Investigating" to "investigating" into the manuscript.

The manuscript has been altered accordingly.

*4.        Line 105. Did the author measure the water content of the dried biomass?*

**Response:**

The water contents of the dried biomass were ~10%.

We have added this information into the manuscript.

The manuscript has been altered to the following:

About 100 g of the dried biomass materials (~10% moisture content) was then introduced into a traditional iron stove commonly used in rural areas (Figure S1).

*5.        Line 150. Please show more details about experimental conditions such as temperature, and pH value of the generated droplets.*

**Response:**

We have showed more details of the droplets experiment in response to the reviewer's suggestion. Temperature and pH information are now included in the main text (Lines 139-144).

In $SO_2$ uptake experiments, the stock solution of BB extracts was premixed with sodium chloride solution (1M) at a volume ratio of 1:1 and the solutions had a pH of 4-6. A droplet generator (Model 201, Uni-Photon Inc.) was then utilized to deposit droplets onto a hydrophobic substrate (model 5793, YSI Inc.) for $SO_2$ uptake experiments. Reactive $SO_2$

uptake experiments were performed via a flow cell/in-situ Raman system at controlled room temperature (23-25°C).

*6.      Line 157. It would be better to give the specific value of the light intensity of xenon lamp.*

**Response:**

The light intensity for our xenon lamp is 1318 mW/cm$^2$.

We have included the information into the manuscript (lines 145-148).

The light experiment was performed using a xenon lamp (model 6258, ozone free, 300W, Newport, light intensity of 1318 mW cm$^{-2}$), with photon flux of 9.8 ×10$^{15}$ photons cm$^{-2}$ s$^{-1}$ in 280-420 nm received by particles in the flow cell (Zhang and Chan, 2023).

*7.      Line 168. The used concentration of bisulfite and BB TOC extracts are 100 ppm and 1 ppm, respectively. Are there environmental reference values for these two concentrations?*

**Response:**

The concentrations of 1 ppm for BB TOC extracts and 100 ppm for bisulfite have significant atmospheric relevance.

Previous studies have indicated that phenolic and non-phenolic aromatic carbonyls from biomass burning plumes can lead to concentrations of up to hundreds of micromolar humid carbonyls (potential PS) in aqueous aerosols, fogs, and clouds (Anastasio et al., 1997). In our study, the 1 ppm TOC concentration aligns well with these atmospheric-relevant ranges and serves as an appropriate representative of aqueous phase photosensitizer concentrations. Furthermore, bisulfite concentrations in ambient cloud and fog water can reach hundreds of micromolar, with total sulfur concentrations (S(IV) + S(VI)) often exceeding several millimolar (Guo et al., 2012; Shen et al., 2012; Rao and Collett, 1995). Therefore, our bisulfite concentration of 100 ppm fits well within the sulfur concentrations detected in cloud and fog water.

We have incorporated this relevant information into our manuscript. (Lines 161-165)

Note that the 1 ppm BB TOC and 100 ppm bisulfite align well with the atmospheric-relevant ranges in aqueous aerosols, fogs and clouds, where PS concentration can reach hundreds of micromolar and total sulfur concentration can exceed several millimolar. (Anastasio et al., 1997; Guo et al., 2012; Shen et al., 2012; Rao and Collett, 1995)

*8.      Line 202. polycyclic aromatic hydrocarbons (PAHs)*

**Response:**

We have added the full name of PAHs for the first appearance.

The manuscript has been altered accordingly.

*9.      Line 204. PAHs.*

**Response:**

The manuscript has been altered accordingly.

*10.      Line 270–271. Did the aging experiments conduct under UV irradiation or OH oxidation condition? The text in lines 121–128 did not show that well, but I think it may be UV irradiation based on descriptions elsewhere, such as line 26 and line 29. If only UV irradiation, 185 nm and 254 nm are not relevant in the troposphere (λ > 290 nm). Please clarify that.*

**Response:**

The UV lamps are used to produce OH• to study OH• oxidation. The oxidation of hydroxyl radicals (OH•) is the predominant oxidation pathway occurring during daylight hours. Irradiation at 185 nm and 254 nm effectively generates OH radicals, leading to the aging of the filter samples, as indicated by the following reactions:

$$H_2O + hv(185 \text{ nm}) \rightarrow OH\bullet + H\cdot \tag{1}$$

$$O_2 + hv(185 \text{ nm}) \rightarrow 2O(^3P) \tag{2}$$

$$O_2 + O(^3P) \rightarrow O_3 \tag{3}$$

$$O_3 + hv(254 \text{nm}) \rightarrow O_2 + O(1D) \tag{4}$$

$$O(^1D) + H_2O \rightarrow 2OH\bullet\cdot \tag{5}$$

$$H\bullet + O_2 \rightarrow HO_2\bullet\cdot \tag{6}$$

The illumination of UV lights creates oxygen atoms, i.e., $O(^3P)$ and $O(^1D)$, which are short-lived and highly reactive with oxygen and water, respectively. The production of ozone in the presence of 185 nm lamp initiates reactions under the presence of 254 nm, which will result in the formation of OH• radicals.

In order to avoid any misunderstanding, we have clarified this in the manuscript.

The manuscript has been altered accordingly.

**Modifications in the manuscript:**

Lines 26-27

The filter aging was achieved by exposing them to OH• through UV irradiation.

Lines 289-291

To investigate the aging effects across various BB materials, we subjected the collected BB filters to OH radical aging by irradiating them with UV lights at wavelengths of 185 nm and 254 nm. This combination effectively generate OH radicals (Tang et al., 2023).

Lins 109-113 To achieve atmospheric OH• aging, the collected fresh BB filter samples were placed in a pre-flushed combustion chamber (zero air, more than 24 h) and illuminated with UV lamps for 40 min. We used lamps of 185 nm and 254 nm, the combination of which have been widely used in oxidation flow reactor design and experiments for mimicking atmospheric OH• concentrations (Peng and Jimenez, 2020; Rowe et al., 2020; Tkacik et al., 2014; Hu et al., 2022).

*11.      Line 283–309. The topic of Sect. 3.2 is aging effects on sulfate formation across various BB materials. The text in line 289–309 provides the explanation about the accelerated sulfate formation in droplets. It would be more in line with the topic to compare whether there is a difference of aging effects between air-water interfaces and bulk phase.*

**Response:**

Thank you for your suggestion.

Air-water interfacial reactions are not the focus of this study, and hence we have decided to remove this discussion to maintain alignment with the main narrative of the manuscript. To address this question, we will need to do experiments with different particle sizes to vary the surface to volume ratio. This would be useful future work.

*12.      Line 316. BrC.*

**Response:**

We have deleted the "brown carbon" in the manuscript.

The manuscript has been altered accordingly.

*13.      Line 349. How to distinguish $WS_{A\text{-}BrC}$ from $WS_A$?*

**Response:**

The method to distinguish $WS_{A\text{-}BrC}$ from $WS_A$ is the same as mentioned before in distinguishing RS and CS (lines 313-318).

$WS_{A\text{-}BrC}$ is a subset of $WS_A$. $WS_A$ and $WS_{A\text{-}BrC}$ are defined as the water-soluble organic species and molecularly identified water-soluble brown carbon in WS extracts, respectively. They can be differentiated based on the criteria raised by Lin et al. (2018). Efficient absorption of visible light in organic molecules necessitates continuous conjugation across a substantial portion of the molecular framework. Compounds exhibiting a Double Bond Equivalent to Carbon (DBE/C) ratio exceeding that of polyenes are promising candidates for brown carbon (BrC) chromophores. For linear polyenes with a general formula of $C_xH_{x+2}$, DBE=0.5×C, while fullerene-like hydrocarbons follow a DBE value of 0.9×C. Compounds with DBE/C ratio higher than polyenes and lower than fullerene-like hydrocarbons can be categorized as potential BrC chromophores. (Lin et al., 2018).

The sentence has been modified to the following to avoid ambiguity (lines 313-318):

Given the presence of chromophoric compounds in BrC (Laskin et al., 2015), we constrained the DBE values to the range of 0.5C≤DBE≤0.9C to semi-qualitatively distinguish BrC

chromophores in the dissolved organic carbon (Lin et al., 2018). $BB_{F/A}$ was defined as the water-soluble organic species while $BB_{F/A\text{-}BrC}$ represented the molecularly identified water-soluble brown carbon falling in the range of $0.5C \leqslant DBE \leqslant 0.9C$ in BB extracts. These definitions will be consistently applied hereafter.

*14.    Line 373. RS extracts.*

**Response:**

We have deleted "rice straw" in the manuscript.

The manuscript has been altered accordingly.

*15.    Line 384. Confirm? More possible explanation should be provided.*

**Response:** We changed the phrase to the following: "On one hand, it supported the findings that chloride participated in the sulfate formation under light but no sulfate formation under dark (Cao et al., 2024; Tang et al., 2023; Zhang and Chan, 2024)."

The manuscript has been altered accordingly.

---

## Author Comment (AC2)

**Point-by-point response to reviewers**

**RC1:**

*Tang et al. reported the enhancement of sulfate production from photosensitization in biomass burning-NaCl droplets. The effects of atmospheric aging and chloride ions in photosensitization-initiated sulfate formation and the corresponding influencing mechanism were also discussed. The experimental results are solid and reliable, but the discussion is not deep enough. I have outlined some of the shortcomings below.*

*In this manuscript, sulfate formation in the atmosphere was investigated by mixing biomass burning and sea-salt particles. My major concern is about the environmental relevant of the designed experiments. The authors showed in detail the reported formation mechanism of sulfates and the prevalence of biomass burning in the Introduction. It is too tedious. The author does not logically explain the importance and necessity of studying sulfate formation by mixing real biomass burning extracts and NaCl. In addition, the relationship between photosensitization and biomass burning should be shown in a few words.*

**Response:** We appreciate the reviewer's thoughtful feedback on our manuscript.

We have shortened our introduction and incorporated the importance and necessity of studying sulfate formation by mixing real BB extracts and NaCl. We also showed the link between photosensitization and biomass burning in the introduction. Overall, when BB air mass mixes with marine air mass, the interactions of BB and sea-salt components are important to atmospheric chemistry, especially multiphase reactions due to the high humidity. The introduction has been modified to the following (lines 36-89):

[revised manuscript text omitted]

The manuscript has been altered accordingly.

Minor comments:

*1.      Please check carefully English statement in the whole manuscript.*

**Response:**

We have conducted a comprehensive check of the manuscript and made relevant modifications to enhance its quality.

*2.      Line 98–100. As mentioned by the authors, the aim of this study includes three points, but the title only covered one point.*

**Response:**

We have modified the title to "Enhanced Sulfate Formation in Mixed Biomass Burning and Sea-salt Interactions Mediated by Photosensitization: Effects of Chloride, Nitrogen-containing Compounds and Atmospheric Aging" to cover all the aspects raised in the abstract.

*3.      Line 100. "investigating".*

**Response:**

We have changed the "Investigating" to "investigating" into the manuscript.

The manuscript has been altered accordingly.

*4.      Line 105. Did the author measure the water content of the dried biomass?*

**Response:**

The water contents of the dried biomass were ~10%.

We have added this information into the manuscript.

The manuscript has been altered to the following:

About 100 g of the dried biomass materials (~10% moisture content) was then introduced into a traditional iron stove commonly used in rural areas (Figure S1).

*5.      Line 150. Please show more details about experimental conditions such as temperature, and pH value of the generated droplets.*

**Response:**

We have showed more details of the droplets experiment in response to the reviewer's suggestion. Temperature and pH information are now included in the main text (Lines 139-144).

In $SO_2$ uptake experiments, the stock solution of BB extracts was premixed with sodium chloride solution (1M) at a volume ratio of 1:1 and the solutions had a pH of 4-6. A droplet generator (Model 201, Uni-Photon Inc.) was then utilized to deposit droplets onto a hydrophobic substrate (model 5793, YSI Inc.) for $SO_2$ uptake experiments. Reactive $SO_2$

uptake experiments were performed via a flow cell/in-situ Raman system at controlled room temperature (23-25°C).

*6.      Line 157. It would be better to give the specific value of the light intensity of xenon lamp.*

**Response:**

The light intensity for our xenon lamp is 1318 mW/cm$^2$.

We have included the information into the manuscript (lines 145-148).

The light experiment was performed using a xenon lamp (model 6258, ozone free, 300W, Newport, light intensity of 1318 mW cm$^{-2}$), with photon flux of $9.8 \times 10^{15}$ photons cm$^{-2}$ s$^{-1}$ in 280-420 nm received by particles in the flow cell (Zhang and Chan, 2023a).

*7.      Line 168. The used concentration of bisulfite and BB TOC extracts are 100 ppm and 1 ppm, respectively. Are there environmental reference values for these two concentrations?*

**Response:**

The concentrations of 1 ppm for BB TOC extracts and 100 ppm for bisulfite have significant atmospheric relevance.

Previous studies have indicated that phenolic and non-phenolic aromatic carbonyls from biomass burning plumes can lead to concentrations of up to hundreds of micromolar humid carbonyls (potential PS) in aqueous aerosols, fogs, and clouds (Anastasio et al., 1997). In our study, the 1 ppm TOC concentration aligns well with these atmospheric-relevant ranges and serves as an appropriate representative of aqueous phase photosensitizer concentrations. Furthermore, bisulfite concentrations in ambient cloud and fog water can reach hundreds of micromolar, with total sulfur concentrations (S(IV) + S(VI)) often exceeding several millimolar (Guo et al., 2012; Shen et al., 2012; Rao and Collett, 1995). Therefore, our bisulfite concentration of 100 ppm fits well within the sulfur concentrations detected in cloud and fog water.

We have incorporated this relevant information into our manuscript. (Lines 161-165)

Note that the 1 ppm BB TOC and 100 ppm bisulfite align well with the atmospheric-relevant ranges in aqueous aerosols, fogs and clouds, where PS concentration can reach hundreds of micromolar and total sulfur concentration can exceed several millimolar. (Anastasio et al., 1997; Guo et al., 2012; Shen et al., 2012; Rao and Collett, 1995)

*8.      Line 202. polycyclic aromatic hydrocarbons (PAHs)*

**Response:**

We have added the full name of PAHs for the first appearance.

The manuscript has been altered accordingly.

*9.      Line 204. PAHs.*

**Response:**

The manuscript has been altered accordingly.

*10.      Line 270–271. Did the aging experiments conduct under UV irradiation or OH oxidation condition? The text in lines 121–128 did not show that well, but I think it may be UV irradiation based on descriptions elsewhere, such as line 26 and line 29. If only UV irradiation, 185 nm and 254 nm are not relevant in the troposphere (λ > 290 nm). Please clarify that.*

**Response:**

The UV lamps are used to produce OH• to study OH• oxidation. The oxidation of hydroxyl radicals (OH•) is the predominant oxidation pathway occurring during daylight hours. Irradiation at 185 nm and 254 nm effectively generates OH radicals, leading to the aging of the filter samples, as indicated by the following reactions:

$$H_2O + hv(185 \text{ nm}) \rightarrow OH• + H· \qquad (1)$$

$$O_2 + hv(185 \text{ nm}) \rightarrow 2O(^3P) \qquad (2)$$

$$O_2 + O(^3P) \rightarrow O_3 \qquad (3)$$

$$O_3 + hv (254nm) \rightarrow O_2 + O(1D) \qquad (4)$$

$$O(^1D) + H_2O \rightarrow 2OH• · \qquad (5)$$

$$H• + O_2 \rightarrow HO_2• · \qquad (6)$$

The illumination of UV lights creates oxygen atoms, i.e., $O(^3P)$ and $O(^1D)$, which are short-lived and highly reactive with oxygen and water, respectively. The production of ozone in the presence of 185 nm lamp initiates reactions under the presence of 254 nm, which will result in the formation of OH• radicals.

In order to avoid any misunderstanding, we have clarified this in the manuscript.

The manuscript has been altered accordingly.

**Modifications in the manuscript:**

Lines 26-27

The filter aging was achieved by exposing them to OH• through UV irradiation.

Lines 289-291

To investigate the aging effects across various BB materials, we subjected the collected BB filters to OH radical aging by irradiating them with UV lights at wavelengths of 185 nm and 254 nm. This combination effectively generate OH radicals (Tang et al., 2023).

Lins 109-113 To achieve atmospheric OH• aging, the collected fresh BB filter samples were placed in a pre-flushed combustion chamber (zero air, more than 24 h) and illuminated with UV lamps for 40 min. We used lamps of 185 nm and 254 nm, the combination of which have been widely used in oxidation flow reactor design and experiments for mimicking atmospheric OH• concentrations (Peng and Jimenez, 2020; Rowe et al., 2020; Tkacik et al., 2014; Hu et al., 2022).

*11.      Line 283–309. The topic of Sect. 3.2 is aging effects on sulfate formation across various BB materials. The text in line 289–309 provides the explanation about the accelerated sulfate formation in droplets. It would be more in line with the topic to compare whether there is a difference of aging effects between air-water interfaces and bulk phase.*

**Response:**

Thank you for your suggestion.

Air-water interfacial reactions are not the focus of this study, and hence we have decided to remove this discussion to maintain alignment with the main narrative of the manuscript. To address this question, we will need to do experiments with different particle sizes to vary the surface to volume ratio. This would be useful future work.

*12.      Line 316. BrC.*

**Response:**

We have deleted the "brown carbon" in the manuscript.

The manuscript has been altered accordingly.

*13.      Line 349. How to distinguish $WS_{A-BrC}$ from $WS_A$?*

**Response:**

The method to distinguish $WS_{A-BrC}$ from $WS_A$ is the same as mentioned before in distinguishing RS and CS (lines 313-318).

$WS_{A-BrC}$ is a subset of $WS_A$. $WS_A$ and $WS_{A-BrC}$ are defined as the water-soluble organic species and molecularly identified water-soluble brown carbon in WS extracts, respectively. They can be differentiated based on the criteria raised by Lin et al. (2018). Efficient absorption of visible light in organic molecules necessitates continuous conjugation across a substantial portion of the molecular framework. Compounds exhibiting a Double Bond Equivalent to Carbon (DBE/C) ratio exceeding that of polyenes are promising candidates for brown carbon (BrC) chromophores. For linear polyenes with a general formula of $C_xH_{x+2}$, $DBE=0.5 \times C$, while fullerene-like hydrocarbons follow a DBE value of $0.9 \times C$. Compounds with DBE/C ratio higher than polyenes and lower than fullerene-like hydrocarbons can be categorized as potential BrC chromophores. (Lin et al., 2018).

The sentence has been modified to the following to avoid ambiguity (lines 313-318):

Given the presence of chromophoric compounds in BrC (Laskin et al., 2015), we constrained the DBE values to the range of $0.5C \leqslant DBE \leqslant 0.9C$ to semi-qualitatively distinguish BrC

chromophores in the dissolved organic carbon (Lin et al., 2018). $BB_{F/A}$ was defined as the water-soluble organic species while $BB_{F/A-BrC}$ represented the molecularly identified water-soluble brown carbon falling in the range of $0.5C \leqslant DBE \leqslant 0.9C$ in BB extracts. These definitions will be consistently applied hereafter.

*14.    Line 373. RS extracts.*

**Response:**

We have deleted "rice straw" in the manuscript.

The manuscript has been altered accordingly.

*15.    Line 384. Confirm? More possible explanation should be provided.*

**Response:** We changed the phrase to the following: "On one hand, it supported the findings that chloride participated in the sulfate formation under light but no sulfate formation under dark (Cao et al., 2024; Tang et al., 2023; Zhang and Chan, 2024)."

The manuscript has been altered accordingly.

**RC2:**

*This study focused on the sulfate formation processes mediated by the mixture of biomass burning extracts and sea salt. Both fresh and photo-aged samples are considered and the sulfate production stories under various scenarios are discussed. This paper can be published on ACP journal, however, before that a major revision is mandatory. Here are some issues for revision.*

**Response:** We appreciate the reviewer's thoughtful feedback and constructive suggestions.

*Major concerns*

*1. This study normalizes sulfate production by TOC. However, the organic and inorganic parts of TOC could be very different, and photosensitization is mainly associated with organic carbons. Authors should explain relevant uncertainties and emphasize the reasonability of such approach.*

**Response:** We acknowledge the reviewer's insightful comment on the use of TOC for normalization.

TOC refers to total organic carbon, which includes the organic photosensitizers (PS). While some inorganic species may also act as PS, the majority of recognized PS are organic compounds. Representative PS in the biomass burning (BB) plume include aromatic carbonyls, non-aromatic carbonyls, and polycyclic aromatic hydrocarbons (PAHs), whose photosensitizing abilities have been confirmed in numerous laboratory studies. (Joo et al., 2024; Zhou et al., 2023; Mabato et al., 2023; Felber et al., 2021; Wang et al., 2020)

Our goal is to compare the photosensitizing ability in different chemical systems, but not to quantify their absolute values. Given the complexity and the lack of a method to quantify PS in BB aerosols, using the total TOC concentration as an upper limit for estimating PS concentration is considered a compromise that allows for systematic comparison. As we have excluded the contribution of reactions other than photosensitization in the BB-NaCl droplet system, the normalization of sulfate formation by TOC yields the lowest estimation of the photosensitizing capacity.

We have added some sentences to justify the TOC normalization method (lines 252-256):

Given the complexity and the lack of a method to quantify PS in BB aerosols, using the total TOC concentration as an upper limit for estimating PS concentration is considered a compromise that allows for systematic comparison. Our goal is to compare the photosensitizing ability in different chemical systems, but not to quantify their absolute values. Therefore, the sulfate formation reported here can be considered as the lower limit of photosensitizing capacity.

*2. The authors perform aqueous experiments along with droplet experiments. Why? I cannot easily get the reasons for this and the readers may also cannot. The authors should explain this and highlight the experimental details (aqueous or droplet) in the related figure captions.*

**Response:** Thank you for your comment.

Aqueous multiphase reactions in the marine boundary layer (MBL) can significantly influence the climate-relevant properties and behavior of marine aerosols. Generally, the matrix of these reactions in the MBL can be categorized into two forms: wet aerosol (aerosol, here in our study as droplets) phase and cloud/fog (bulk aqueous) phase. The reaction rates were highly related to the intrinsic microphysical and chemical nature of the reaction matrix (Ruiz-Lopez et al., 2020; Herrmann, 2003). While the reaction temperature ranges were the same for aqueous phase and droplet reactions, the aerosol droplet reactions were characterized by high ionic strength (up to >10 M), low liquid water content ($10^{-7}$-$10^{-3}$ cm$^3$ m$^{-3}$) and high surface-to-volume ratio. However, the bulk aqueous reactions were characterized by low ionic strength ($10^{-5}$ to $10^{-2}$ M), high liquid water content ($10^{-2}$ cm$^3$ m$^{-3}$) and low surface-to-volume ratio. (Herrmann et al., 2015). Due to the differences in liquid water contents, ionic strength and chemical composition concentrations as well as the surface-to-volume ratio, the reaction rate constants towards droplet and aqueous reactions may be totally different. Hence, it is necessary to study both droplet and cloud(bulk) phase reactions to understand MBL chemistry.

Various experimental tools have been employed to investigate reactions under aerosol and cloud conditions. To study aerosol phase reactions, researchers have utilized microreactors—such as microdroplets generated through various electrospray and spray methods, levitated droplets, thin films on surfaces, and microfluidic systems. To study aqueous phase reactions, aqueous reactors, also known as aqueous phase simulation chambers, are commonly used to study rate constants and the degradation or conversion of chemical species in bulk aqueous solutions.

We have incorporated the discussions of doing such experiments in introduction part. (lines 36-89)

Since the sulfate formation rate depends on the intrinsic properties of the solution matrix and the two main reaction matrixes in marine boundary layer (MBL) were wet aerosol (droplet in our case) and cloud/fog (bulk aqueous), both droplet and aqueous reactions are relevant for studying the aqueous reactions in aerosols and clouds within MBL (Ruiz-Lopez et al., 2020; Herrmann, 2003). Typically, droplet experiments were characterized by high ionic strength (up to >10 M), low liquid water content ($10^{-7}$-$10^{-3}$ cm$^3$ m$^{-3}$) and high surface-to-volume ratio whereas aqueous reactions exhibit the opposite characteristics. Transmission electron microscopy (TEM) studies indicate that most coastal particles are internally mixed, showing a higher proportion of organic and salt mixtures in the presence of biomass burning aerosols, accompanied by an increase in sulfate (Dang et al., 2022; Li et al., 2003) However, discrepancies persist between modeled simulations and measured sulfate concentrations in MBL (Yu et al., 2023). The interactions of sea-salt and BB aerosols, especially in multiphase reactions, can potentially unravel the intricate chemistry of sulfate formation in BB affected MBL. Hence, internal mixtures of inorganic salt and water-soluble organic carbons are often used in reaction studies (Tan et al., 2024).

The experimental details (aqueous or droplet) have been highlighted in the related figure captions (Figs. 1-4).

*3.     The interesting results of Figure S3 are suggested to be shown in main text. But it depends on the authors. Moreover, replacement of air by $N_2$ in distinguishing direct or indirect photosensitization reactions should be introduced with more details.*

**Response:** Thank you for your kind suggestions.

We have moved the Fig. S3 in the supplementary information to Fig. 2 in the manuscript.

We elaborated the replacement of air by $N_2$ by adding the following sentences into the manuscript (lines 266-268):

Under $N_2$-saturated conditions, secondary oxidants such as $HO_2\bullet$, $OH\bullet$ oxidation pathway can be ruled out due to the lack of oxygen. Consequently, the sulfate formed under this condition can be considered as the direct PS* oxidation.

*4.     The authors discuss the variations of molecular characteristics (based on MS techniques) and their effects on the oxidation event. Anyway, the relevant contents are much more complex than I expect and you should slightly simplify these parts and also show the different molecular characteristics before and after aging. How about the possible quantitative association between these characteristics and the oxidation kinetics?*

**Response:** Thank you for your thoughtful suggestions.

As indicated by the Pearson correlation heatmap (Figure S9), we observed a strong correlation between sulfate formation rates and the CHO, CHON, and CHN species. In our manuscript, we elaborate on how the proportions of these species change, acknowledging that their variations are not uniform across different BB materials. We believe this nuanced discussion enhances the understanding of complex relationships, especially for readers interested in detailed MS analysis.

We have dedicated additional paragraphs to explore the differences in chemical species associated with different BB materials. Specifically, we tentatively identified CHON-species as nitrophenols and CHN+ species as N-heterocyclic alkaloids, with potential formulas of $C_5H_6N_2(CH_2)_n$, $C_5H_8N_2(CH_2)_n$, and $C_7H_6N_2(CH_2)_n$.

For the CHO species, we applied various criteria to identify potential photosensitizers, such as the aromatic index, maximum carbonyl ratio, H/C and O/C ratios. While we acknowledge that precise molecular structures cannot be determined solely from elemental compositions, we believe our discussions provide meaningful insights into the chemical behavior and potential reactivity of these species.

To summarize, we propose that the enhanced sulfate formation in $CS_A$ and $RS_A$ was likely due to the increased proportions (by intensity) of CHON and CHN species, potentially nitrophenols and N-heterocyclic compounds. Conversely, the increased sulfate formation in $WS_A$ appears to be linked to a higher percentage of CHO species. Future research can be performed to identify more detailed molecular characteristics.

We have attempted to explore the associations between chemical characteristics and sulfate formation potential. However, the correlation coefficients obtained from both linear and

non-linear fittings were found to lack scientific meaning, as they were heavily influenced by the proportions of the species involved.

Given these complexities, we concluded that such quantitative relationships are not universally applicable across all samples. Therefore, we decided not to include these data in the manuscript to avoid unintended misinterpretations. Nevertheless, we added a sentence to explain the lack of such discussion in the paper: However, the associations between detailed chemical characteristics and sulfate formation were not provided in this study due to the complexity of the interactions between different chemical categories and difficulties in the interpretation of the coefficients. Future studies are needed to elucidate the relationships between sulfate formation and the chemical characteristics.

We appreciate your understanding and hope this clarification addresses your concerns.

We have added more sentences to summarize the MS results in the manuscript (lines 372-379.

To summarize, we propose that the enhanced sulfate formation in $CS_A$ and $RS_A$ was likely due to the increased proportions (by intensity) of CHON and CHN species, potentially nitrophenols and N-heterocyclic compounds. Conversely, the increased sulfate formation in $WS_A$ appears to be linked to a higher percentage of CHO species. However, the associations between detailed chemical characteristics and sulfate formation were not provided in this study due to the complexity of the interactions between different chemical categories and difficulties in the interpretation of the coefficients. Future studies are needed to elucidate the relationships between sulfate formation and the chemical characteristics.

5. . *The droplet and aqueous-phase experiments were performed under simulated sunlight. The paper would be more attractive if the comparison between dark and irradiation experiments can be conducted, at least in one section. Especially for section 3.3 in my opinion because the effect of ionic strength should be emphasized.*

**Response:** Thank you for your comment.

In fact, we do have dark and irradiation comparison in our work. As we mentioned in Section 3.1 "The absence of sulfate formation under dark conditions ruled out the involvement of direct $H_2O_2$ oxidation and organic peroxide oxidation pathways". Specifically, we did not find any sulfate formation under dark for all the BB-NaCl droplet experiments. Hence, we put emphasis on light experiments with little discussions on dark reactions. Nevertheless, we elaborated in the experimental section for better clarity.

Section 2.3 (lines 148-149)

Identical experiments were conducted in the dark, with the lights off and the experimental area kept in complete darkness.

Section 3.1 (lines 173-174)

As no sulfate was detected in the dark conditions for any of the experiments, we have focused on the light experiments.

Minor concerns

1. *Line 106. Please provide a picture of iron stove.*

**Response:**

Figure S1 illustrates the design of the iron stove we used.

[Figure]

Figure S1 Image of the iron stove used in this study.

The relevant figure has been attached to the supporting information (revised supporting information Fig. S1).

2. *It would be better to denote the "aging" process as "photo-aging"*

**Response:**

The UV lamps are used to produce OH• to study OH• oxidation. The oxidation of hydroxyl radicals (OH•) is the predominant oxidation pathway occurring during daylight hours. Irradiation at 185 nm and 254 nm effectively generates OH radicals, leading to the aging of the filter samples, as indicated by the following reactions:

$$H_2O + hv(185 \text{ nm}) \rightarrow OH\bullet + H\cdot \tag{1}$$

$$O_2 + hv(185 \text{ nm}) \rightarrow 2O(^3P) \tag{2}$$

$$O_2 + O(^3P) \rightarrow O_3 \tag{3}$$

$$O_3 + hv\ (254\text{nm}) \rightarrow O_2 + O(1D) \tag{4}$$

$$O(^1D) + H_2O \rightarrow 2OH\bullet\cdot \tag{5}$$

$$H\bullet + O_2 \rightarrow HO_2\bullet\cdot \tag{6}$$

The illumination of UV lights creates oxygen atoms, i.e., O($^3$P) and O($^1$D), which are short-lived and highly reactive with oxygen and water, respectively. The production of ozone in the presence of 185 nm lamp initiates reactions under the presence of 254 nm, which will result in the formation of OH• radicals.

In order to avoid any misunderstanding, we have clarified this in the manuscript.

The manuscript has been altered accordingly.

**Modifications in the manuscript:**

Lines 26-27.

The filter aging was achieved by exposing them to OH• through UV irradiation.

Lines 289-291

To investigate the aging effects across various BB materials, we subjected the collected BB filters to OH radical aging by irradiating them with UV lights at wavelengths of 185 nm and 254 nm. This combination effectively generate OH radicals (Tang et al., 2023).

Lins 109-113

To achieve atmospheric OH• aging, the collected fresh BB filter samples were placed in a pre-flushed combustion chamber (zero air, more than 24 h) and illuminated with UV lamps for 40 min. We used lamps of 185 nm and 254 nm, the combination of which have been widely used in oxidation flow reactor design and experiments for mimicking atmospheric OH• concentrations (Peng and Jimenez, 2020; Rowe et al., 2020; Tkacik et al., 2014; Hu et al., 2022)

*3.    I don't think incense is one kind of biomass.*

**Response:** Thank you for your comment.

To the best of our knowledge, there is no definitive composition of biomass burning as the biomass varies a lot in composition. Wood powder is one of the ubiquitous components in incense sticks. Wood is clearly a biomass and therefore, burning incense, in a general sense, should be considered one type of biomass burning, although it may not be forest fires, wood stove burning etc., whose chemical composition also depends on the type of vegetation and wood burned. We reckon that they are all generally called biomass burning, without well-defined parametric constraints on properties. Hence, we believe that it is logical to call incense burning biomass burning too. Nevertheless, we provide more scientific discussion and justification below on relating incense burning and biomass burning.

There are previous reports on the resemblance of the particulate mass spectra between incense burning and biomass burning aerosols. (Yang et al., 2017; Li et al., 2012) For

example, Li et al. (2012) compared the UMR mass spectra from incense burning experiments with those from other BBOA by correlations of BBOA spectra with three representative mass spectra of biomass nature. Results showed that the mass spectra from incense burning particles correlate well (Table R1) with both mass spectra of lignin powder and PMF-resolved BBOA, suggesting their similarities in chemical characteristics.

Table R1: Correlation coefficients (R) between the mass spectra from mosquito coil and levoglucosan, lignin powder and PMF-resolved BBOA (reconstructed from (Li et al., 2012))

| Correlation Coefficient | Levoglucosan | Lignin powder | PMF-resolved BBOA |
|---|---|---|---|
| Mosquito Coil | 0.86 | 0.89 | 0.98 |

To further verify the molecular composition similarities between incense burning (IS) and biomass-burning particles, we collected fresh burning particles from 9 kinds of incense with different brands, origins, and usages and three typical biomass including wheat, rice stalk, corn straw, and eucalyptus branches on 47-mm quartz filters. The collected incense-burning and biomass-burning samples were analyzed using comprehensive two-dimensional gas chromatography-mass spectrometer (GC×GC-MS, GC-MS TQ 8050, SHIMADZU) coupled with a thermal desorption system (TDS 3, C 506, GERSTEL). Chromatograms of IS and biomass-burning particles were compared by pixel-based partial least squares-discriminant analysis (PLS-DA) conducted in R program language. (Quiroz-Moreno et al., 2020) Chromatograms were read in network common data form (NetCDF) and then smoothed and baseline-corrected. Two-dimensional correlation optimized warping (2DCOW) was utilized to align chromatograms. (Song et al., 2022) The aligned chromatograms, which were sub grouped into "incense" and "biomass" for further PLS-DA analysis, were unfolded and then introduced to the mixomics package. (González et al., 2012)

Typical GC×GC chromatograms of incense-burning and biomass-burning particles are exhibited in Figure R1. Significant similarities among chromatograms are observed with high contributions of sugars (levoglucosan) and phenols. Pixel-based PLS-DA was

conducted by subgrouping GC×GC chromatograms into incense and biomass combustion (Figure R2). Although chromatograms are subgrouped, the separation of PLS-DA still reveals that there is no systematic difference among samples. In other words, similarities between IS and biomass burning are greatly larger than differences from a pixel-based level, highlighting the surrogate of IS to biomass burning.

From the elution patterns of GC×GC chromatograms and the pixel-based PLS-DA results, we can conclude that the incense-burning particles investigated in this work are representative irrespective of the shape and material of incenses and can be utilized for surrogates of biomass-burning emissions.

Detailed information has been provided in our previous work. (Tang et al., 2023)

[Figure]

Figure R1. Typical GC×GC chromatogram of: (a) incense-burning particles; (b) biomass-burning particles

[Figure]

Figure R2. Pixel-based PLS-DA analysis of GC×GC chromatogram of incense- and biomass-burning particles.

The sentence has been changed to the following (lines 104-105):

Our previous study demonstrated that IS was representative of BB based on GC×GC chromatograms and pixel-based partial least squares discriminant analysis. (Tang et al., 2023)

*4.  The light wavenumbers were 185 nm and 254 nm, both of which were much lower than the common range of sunlight in the real atmosphere.*

**Response:** Thank you pointing out that.

We have modified the manuscript to make it clear that we use 185 nm and 254 nm UV lights to provide the OH atmosphere to simulate the atmospheric OH aging. Details can be seen in the response to question 2.

*5.  Line 169. Please explain "to achieve air-saturated conditions"*

**Response:**

In our study, "air-saturated conditions" refer to the state in which a liquid solution contains the saturated amount of dissolved air (typically oxygen and nitrogen) that can be held at a given temperature and pressure. Achieving air saturation is crucial for our experiments as

it ensures that the solutions are representative of natural environmental conditions, where physiochemical and biochemical processes are exposed to air-saturated water.

To attain these conditions, we continuously introduced synthetic air into the solutions at a flow rate of 0.5 L min⁻¹. This flow rate was chosen to provide enough air to saturate the solution without causing excessive turbulence, which could disrupt the system or lead to inaccurate measurements.

6.      *Some supplementary figures (e.g. Figure S8) are not discussed in main text.*

**Response:**

We have indeed discussed Figure S8 in Section 3.2 of the main text, specifically regarding the statement, "Approximately 80% of the CHN+ species…benzimidazole or indazole." To avoid any misunderstanding, we have now marked Figure S8 in the relevant position.

7.      *The bottom part of Table 1 is not clear.*

**Response:**

It has been changed to "ᵃThe $n\gamma_{SO_2}$ was calculated by normalizing the $\gamma_{SO_2}$ with the TOC concentration in the BB extracts, i.e., $n\gamma_{SO_2} = \gamma_{SO_2}/TOC$."

8.      *Line 309. "out of the scope of this study." The authors are suggested to simplify some parts of the paper to make the text more readable.*

Response:

We have deleted the discussions on the accelerated sulfate formation rate of droplets compared to the aqueous reactions to make the story clearer.

9.      *Do you have literatures supporting "maximum carbonyl ratio (MCR)"?*

**Response:**

Indeed, there is literature supporting the concept of the "maximum carbonyl ratio" (MCR).

The term "MCR" was introduced by Zhang et al. (2021b) and is derived from molecular composition analyses using UHPLC-Orbitrap MS for identifying and characterizing organic aerosols. This concept has been demonstrated through analyses of ambient organic aerosol (OA) samples from both urban settings (central Beijing) and forested areas (Hyytiälä, boreal forest), as well as laboratory-generated secondary organic aerosol (SOA).

Since then, additional studies have utilized MCR as an indicator to predict the potential maximum number of carbonyl groups in various aerosols (Wang et al., 2024; Calderon-Arrieta et al., 2024; Liu et al., 2023). CHO species that fall within the region of MCR ⩾ 0.9 are associated with oxidized unsaturated and highly unsaturated compounds such as typical PS of imidazole-carboxaldehyde and polycyclic aromatic hydrocarbons (PAHs).

We have incorporated more references into the manuscript and explained the reason of using MCR⩾0.9 as a screening value.

The manuscript has been altered to the following (lines 354-358 and 363-368):

Therefore, we filtered the chemical formula of CHO- species from UHPLC-Orbitrap-HRMS by applying the maximum carbonyl ratio (MCR) (Zhang et al., 2021b; Wang et al., 2024; Calderon-Arrieta et al., 2024; Liu et al., 2023), H/C, O/C as well as modified aromaticity index ($AI_{mod}$) to focus on potential PS (Zherebker et al., 2022; Koch and Dittmar, 2006).

By applying a data filtration process involving CHO-, condensed aromatics, polyphenolics, highly unsaturated and phenolic compounds based on the aforementioned criteria, as well as MCR≥0.9 (which includes oxidized unsaturated and highly unsaturated compounds such as PS like imidazole-carboxaldehyde and PAHs) (Zhang et al., 2021b), 52.6% and 49.7% of the compounds (by intensity) can be considered as potential PS in $WS_A$ and $WS_F$, respectively.

*10.     Figure S7. How about the variation of water acidity as you measured both sulfate and bisulfate.*

**Response:**

As the experiment proceeded, sulfate concentrations accumulated while bisulfite concentrations decreased. Concurrently, the pH of the aqueous solution decreased from approximately 5.0 to 3.0, reflecting enhanced acidity.

We have incorporated the sentence above into the manuscript (lines 304-306).

*11.     Please add error bars to Figure 3a.*

**Response:**

The error bars have been added to Figure 3a.

[Figure]

*12.    The authors are suggested to kinetically compare the pathway they study with those documented.*

**Response:**

We have incorporated the comparison between the reactive SO₂ uptake coefficients with those documented in the previous literature. Please kindly find the following sentences. (lines 211-226)

The $\gamma_{SO_2}$ obtained in our study are $0.9 - 6.6 \times 10^{-6}$, which are consistent but fall on the low side of the reported heterogeneous SO₂ oxidation processes, including nitrate photolysis ($10^{-6}$-$10^{-5}$) (Gen et al., 2019), TMI-catalyzed oxidation ($10^{-6}$-$10^{-4}$) (Zhang et al., 2024a), NO₂/O₃ oxidation ($10^{-6}$-$10^{-4}$) (Zhang et al., 2021a; Zhang and Chan, 2023b) and peroxide oxidation ($10^{-6}$-$10^{-1}$) (Wang et al., 2021; Ye et al., 2018; Yao et al., 2019). Additionally, the reported $\gamma_{SO_2}$ in our study aligns well with the results obtained from ambient samples in Beijing (Zhang et al., 2020). The large discrepancy of the reported $\gamma_{SO_2}$ can be attributed to the differences in aerosol components, particle size, RH, SO₂ and oxidants concentrations. From our results, it appears that sulfate formation from BB-NaCl particles is much less effective than particles under nitrate photolysis. It is interesting to note that Zhou et al. (2023) found particles coated with model PS compounds much more effective in sulfate formation than nitrate particles under photolysis in a PAM reactor. The much shorter residence time in that reactor (2.5 min) and higher PS concentration (~66 mM) than the exposure time of filter samples (40 min) and PS concentration (<250 ppm) in our sulfate experiments may explain the differences in the comparison of PS/BB and nitrate photolysis results.

The manuscript has been altered accordingly.

**RC3**

*Tang et al.'s study explored sulfate formation by mixing real biomass burning (BB) extracts with NaCl, simulating internal mixtures of BB and sea-salt particles. The research highlighted that photosensitizers present in real BB particles can significantly enhance sulfate production. Additionally, potential photosensitizers in BB particles were tentatively identified at the molecular level. Overall, the paper is well-written, and I have only a few minor comments for consideration.*

**Response**: We appreciate the reviewer's endorsement of our manuscript.

*Comments:*

*1.      Lines 41-52: The authors discuss newly identified pathways for aerosol sulfate formation while acknowledging notable discrepancies between model simulations and observed sulfate concentrations. However, the cited studies (Wyant et al. 2015 and Wang et al. 2014) predate these findings, which raises questions about their relevance. Even with these new pathways included, do gaps between modeled simulations and measured sulfate concentrations still remain?*

**Response**: Thank you for your thoughtful comments.

We agree that the work we cited predates the new findings of sulfate formation mechanisms. Therefore, we have updated the cited work and rewrote the introduction part as follows (lines 36-89)

[revised manuscript text omitted]

Regarding if the gaps remain between modeled simulations and observed concentrations after incorporating the new formation pathway into the model, the answer is both yes and no. Integrating new models into the sulfate formation framework can certainly increase the modeled concentrations and reduce the discrepancy between modeled and measured sulfate levels. However, the extent to which this can be achieved is still an open question.

On one hand, the parameterization of different processes such as $NO_2$ oxidation, TMI-catalyzed oxidation, and nitrate photolysis is ongoing and exhibits varying performances across different studies. (Zhang et al., 2024a; Zhang et al., 2024b; Gao et al., 2024). Even for the traditional pathways like $SO_2$ oxidation by $O_3$, there are still new findings. For example, Yu et al. (2023) discovered that the ionic strength can enhance the sulfate formation rate of $SO_2$ by $O_3$ under cloud-free conditions in the MBL. On the other hand, several factors related to photosensitization processes remain inadequately addressed. For instance, Wang et al. (2020) highlighted the significance of photosensitization in Beijing's winter haze. The sulfate production rates can vary widely, with the highest estimated value (7.9 μg m$^{-3}$ h$^{-1}$) even surpassing the ambient sulfate formation rate. This variability can be attributed to uncertainties in photosensitizer concentrations, the quantum yield of triplet-state photosensitizers, and the formation of secondary reactive oxygen species through photosensitizer-mediated reactions.

In mixed systems, as indicated by our group's previous work (Mabato et al., 2024), photosensitizers can act both as reactants and oxidants, complicating the evolutionary processes observed in real ambient samples. Therefore, a comprehensive resolution is not achievable in a single step. There is still considerable work ahead to establish a closer alignment between measured and simulated sulfate levels. In this work, we examined the potential contributions of different real BB particles in sulfate formation and we highlight the importance of photosensitization reactions.

*2.       Lines 123-125: While some prior studies have used 185 nm and 254 nm lamps for irradiation, I believe this method does not adequately replicate real-world conditions. The composition of the irradiated BB samples in the study may not accurately reflect the aging process of actual BB particles.*

 **Response**: Thank you for your valuable feedback regarding the use of 185 nm and 254 nm lamps in our study.

As we mentioned in the manuscript, the 185 nm and 254 nm lamps are specifically chosen to generate OH radicals, creating an OH• oxidative environment that simulates the aging of biomass burning (BB) particles.

We acknowledge that the oxidation processes of biomass particles in the real atmosphere are indeed more complex than those we can replicate in the laboratory. This limitation is a common challenge in many atmospheric studies, as it is difficult to capture all aspects of atmospheric conditions. However, we aimed to create a representative scenario to reflect OH• aging in our experiments using UV lamps, which have been accepted as a source of OH radicals in the literature.

We are continually working to strengthen our research and explore additional aspects of biomass burning aging, which is complex due to variability of the chemical compositions of the BB particles and the different potential oxidation mechanisms. In this work, we identified the role of photosensitization reactions in sulfate formation in BB particles.

We have modified the description in our manuscript to clarify our methodology as follows:

Lines 109-113

To achieve atmospheric OH• aging, the collected fresh BB filter samples were placed in a pre-flushed combustion chamber (zero air, more than 24 h) and illuminated with UV lamps for 40 min. We used lamps of 185 nm and 254 nm, the combination of which have been widely used in oxidation flow reactor design and experiments for mimicking atmospheric OH• concentrations (Peng and Jimenez, 2020; Rowe et al., 2020; Tkacik et al., 2014; Hu et al., 2022).

Lines 289-290

To investigate the aging effects across various BB materials, we subjected the collected BB filters to OH radical aging by irradiating them with UV lights at wavelengths of 185 nm and 254 nm. This combination effectively generate OH radicals (Tang et al., 2023).

Thank you again for your constructive comments.

Anastasio, C., Faust, B. C., and Rao, C. J.: Aromatic Carbonyl Compounds as Aqueous-Phase Photochemical Sources of Hydrogen Peroxide in Acidic Sulfate Aerosols, Fogs, and Clouds. 1. Non-Phenolic Methoxybenzaldehydes and Methoxyacetophenones with Reductants (Phenols), Environmental Science & Technology, 31, 218-232, 10.1021/es960359g, 1997.

Calderon-Arrieta, D., Morales, A. C., Hettiyadura, A. P. S., Estock, T. M., Li, C., Rudich, Y., and Laskin, A.: Enhanced Light Absorption and Elevated Viscosity of Atmospheric Brown Carbon through Evaporation of Volatile Components, Environmental Science & Technology, 58, 7493-7504, 10.1021/acs.est.3c10184, 2024.

Cao, Y., Liu, J., Ma, Q., Zhang, C., Zhang, P., Chen, T., Wang, Y., Chu, B., Zhang, X., Francisco, J. S., and He, H.: Photoactivation of Chlorine and Its Catalytic Role in the Formation of Sulfate Aerosols, Journal of the American Chemical Society, 146, 1467-1475, 10.1021/jacs.3c10840, 2024.

Felber, T., Schaefer, T., He, L., and Herrmann, H.: Aromatic Carbonyl and Nitro Compounds as Photosensitizers and Their Photophysical Properties in the Tropospheric Aqueous Phase, The Journal of Physical Chemistry A, 125, 5078-5095, 10.1021/acs.jpca.1c03503, 2021.

Gao, J., Wang, H., Liu, W., Xu, H., Wei, Y., Tian, X., Feng, Y., Song, S., and Shi, G.: Hydrogen peroxide serves as pivotal fountainhead for aerosol aqueous sulfate formation from a global perspective, Nature Communications, 15, 4625, 2024.

Gen, M., Zhang, R., Huang, D. D., Li, Y., and Chan, C. K.: Heterogeneous SO2 Oxidation in Sulfate Formation by Photolysis of Particulate Nitrate, Environmental Science & Technology Letters, 6, 86-91, 10.1021/acs.estlett.8b00681, 2019.

González, I., Cao, K.-A. L., Davis, M. J., and Déjean, S.: Visualising associations between paired 'omics' data sets, BioData mining, 5, 1-23, 2012.

Guo, J., Wang, Y., Shen, X., Wang, Z., Lee, T., Wang, X., Li, P., Sun, M., Collett, J. L., Wang, W., and Wang, T.: Characterization of cloud water chemistry at Mount Tai, China: Seasonal variation, anthropogenic impact, and cloud processing, Atmospheric Environment, 60, 467-476, https://doi.org/10.1016/j.atmosenv.2012.07.016, 2012.

Herrmann, H.: Kinetics of aqueous phase reactions relevant for atmospheric chemistry, Chemical reviews, 103, 4691-4716, 2003.

Herrmann, H., Schaefer, T., Tilgner, A., Styler, S. A., Weller, C., Teich, M., and Otto, T.: Tropospheric Aqueous-Phase Chemistry: Kinetics, Mechanisms, and Its Coupling to a Changing Gas Phase, Chemical Reviews, 115, 4259-4334, 10.1021/cr500447k, 2015.

Hu, W., Zhou, H., Chen, W., Ye, Y., Pan, T., Wang, Y., Song, W., Zhang, H., Deng, W., Zhu, M., Wang, C., Wu, C., Ye, C., Wang, Z., Yuan, B., Huang, S., Shao, M., Peng, Z., Day, D. A., Campuzano-Jost, P., Lambe, A. T., Worsnop, D. R., Jimenez, J. L., and Wang, X.: Oxidation Flow Reactor Results in a Chinese Megacity Emphasize the Important Contribution of S/IVOCs to Ambient SOA Formation, Environmental Science & Technology, 56, 6880-6893, 10.1021/acs.est.1c03155, 2022.

Joo, T., Machesky, J. E., Zeng, L., Hass-Mitchell, T., Weber, R. J., Gentner, D. R., and Ng, N. L.: Secondary Brown Carbon Formation From Photooxidation of Furans From Biomass Burning, Geophysical Research Letters, 51, e2023GL104900, https://doi.org/10.1029/2023GL104900, 2024.

Koch, B. P. and Dittmar, T.: From mass to structure: An aromaticity index for high-resolution mass data of natural organic matter, Rapid communications in mass spectrometry, 20, 926-932, 2006.

Laskin, A., Laskin, J., and Nizkorodov, S. A.: Chemistry of Atmospheric Brown Carbon, Chemical Reviews, 115, 4335-4382, 10.1021/cr5006167, 2015.

Li, Y. J., Yeung, J. W., Leung, T. P., Lau, A. P., and Chan, C. K.: Characterization of organic particles from incense burning using an aerodyne high-resolution time-of-flight aerosol mass spectrometer, Aerosol science and technology, 46, 654-665, 2012.

Liang, Z., Li, Y., Go, B. R., and Chan, C. K.: Complexities of Photosensitization in Atmospheric Particles, ACS ES&T Air, 10.1021/acsestair.4c00112, 2024.

Lin, P., Fleming, L. T., Nizkorodov, S. A., Laskin, J., and Laskin, A.: Comprehensive Molecular Characterization of Atmospheric Brown Carbon by High Resolution Mass Spectrometry with Electrospray and Atmospheric Pressure Photoionization, Analytical Chemistry, 90, 12493-12502, 10.1021/acs.analchem.8b02177, 2018.

Liu, D., Zhang, Y., Zhong, S., Chen, S., Xie, Q., Zhang, D., Zhang, Q., Hu, W., Deng, J., Wu, L., Ma, C., Tong, H., and Fu, P.: Large differences of highly oxygenated organic molecules (HOMs) and low-volatile species in secondary organic aerosols (SOAs) formed from ozonolysis of β-pinene and limonene, Atmos. Chem. Phys., 23, 8383-8402, 10.5194/acp-23-8383-2023, 2023.

Mabato, B. R. G., Li, Y. J., Huang, D. D., Wang, Y., and Chan, C. K.: Comparison of aqueous secondary organic aerosol (aqSOA) product distributions from guaiacol oxidation by non-phenolic and phenolic methoxybenzaldehydes as photosensitizers in the absence and presence of ammonium nitrate, Atmos. Chem. Phys., 23, 2859-2875, 10.5194/acp-23-2859-2023, 2023.

Peng, Z. and Jimenez, J. L.: Radical chemistry in oxidation flow reactors for atmospheric chemistry research, Chemical Society Reviews, 49, 2570-2616, 2020.

Quiroz-Moreno, C., Furlan, M. F., Belinato, J. R., Augusto, F., Alexandrino, G. L., and Mogollón, N. G. S.: RGCxGC toolbox: An R-package for data processing in comprehensive two-dimensional gas chromatography-mass spectrometry, Microchemical Journal, 156, 104830, 2020.

Rao, X. and Collett, J. L. J.: Behavior of S (IV) and formaldehyde in a chemically heterogeneous cloud, Environmental science & technology, 29, 1023-1031, 1995.

Rowe, J. P., Lambe, A. T., and Brune, W. H.: Technical Note: Effect of varying the $\lambda$ = 185 and 254 nm photon flux ratio on radical generation in oxidation flow reactors, Atmos. Chem. Phys., 20, 13417-13424, 10.5194/acp-20-13417-2020, 2020.

Ruiz-Lopez, M. F., Francisco, J. S., Martins-Costa, M. T. C., and Anglada, J. M.: Molecular reactions at aqueous interfaces, Nature Reviews Chemistry, 4, 459-475, 10.1038/s41570-020-0203-2, 2020.

Shen, X., Lee, T., Guo, J., Wang, X., Li, P., Xu, P., Wang, Y., Ren, Y., Wang, W., Wang, T., Li, Y., Carn, S. A., and Collett, J. L.: Aqueous phase sulfate production in clouds in eastern China, Atmospheric Environment, 62, 502-511, https://doi.org/10.1016/j.atmosenv.2012.07.079, 2012.

Song, K., Gong, Y., Guo, S., Lv, D., Wang, H., Wan, Z., Yu, Y., Tang, R., Li, T., and Tan, R.: Investigation of partition coefficients and fingerprints of atmospheric gas-and particle-phase intermediate volatility and semi-volatile organic compounds using pixel-based approaches, Journal of Chromatography A, 1665, 462808, 2022.

[revised manuscript text omitted]

---

## Author Response (AR2)

**Point-by-point response to reviewers**

**Comments**

1.      "The authors claim that the performance of droplet and bulk experiments follows natural rules in the MBL involving aerosol and cloud particles. However, this may not be entirely convincing, as droplet experiments can also simulate low ionic strength conditions. Some interfacial characteristics might be overlooked in bulk reactions, and I believe the authors should discuss these limitations (beyond just droplet vs. bulk tests) in the revised paper.
**Response:** We thank the reviewer for the comment.
We have revised our manuscript to clarify this (lines 74-76 and lines 311-318):
"Additionally, droplet experiments can encompass certain interfacial reaction pathways that may occur in atmospheric conditions."

"Lower sulfate formation rates were observed for bulk reactions compared to droplets reactions, which may be attributed to the accelerated reactions induced by PS at the air-water interface, as well as differences in concentrations of S(IV) and NaCl. However, given that interfacial reactions are closely linked to particle size (Wei et al., 2020; Chen et al., 2022b), and additional research is needed to better understand its influence. Our experiments involve large droplets of the size of 60μm. The interfacial effects of such large droplets may not be evident. Future work should use submicron and nanometer size particles to examine the interfacial effects."
The manuscript has been altered accordingly.

2.      The discussion on direct and indirect photosensitization reactions is somewhat disorganized. In my view, some molecular oxygen may be present in the commercial or home-made HULIS samples, which could also be a limitation that should be emphasized. Additionally, the role of Ar as a carrier gas was not considered.
**Response:**

Some molecular oxygen may be present in the extracted BB and IS samples. However, its concentration is likely low under $N_2$-saturated conditions, as indicated by the absence of sulfate in the controlled experiments of IS-NaCl droplets. Hence, even if they are present, the molecular oxygen did not seem to play any appreciable role. We will address this point in our revised discussion.

Lines 268-271

"Despite initial molecular oxygen in the droplets may also participate in sulfate formation under $N_2$-saturated conditions, its contributions are likely minimal. Therefore, the sulfate formed under $N_2$-saturated condition can be considered as the upper limit of direct $^3PS^*$ oxidation."

Regarding the use of argon as a carrier gas, we consider its role similar to that of nitrogen, both providing an inert atmosphere; thus, we did not conduct experiments specifically with argon.

3.      Why did the authors choose Pearson correlation analysis over Spearman correlation analysis? Was the distribution of the target dataset tested?

**Response:** Thanks for your comment.

We conducted the Shapiro-Wilk test to assess the normality of the dataset, as shown in Table S4. A Shapiro-Wilk test result of p > 0.05 indicates that the dataset can be considered normally distributed, whereas p⩽0.05 suggests non-normality. Given these considerations, Spearman correlation analysis is more appropriate for the heatmap analysis, while Pearson analysis is more suitable for specific parameters such as $k_{SO4}$, nitrate, CHN+ and others+. Consequently, we have revised Fig. S9 to Spearman correlation heatmap. The manuscript has been modified to the following:

(Lines 424-426)

"Statistical analysis using the Spearman correlation coefficients, as guided by the Shapiro-Wilk test (Table S4), revealed that the CHO, CHON, and CHN species exhibited significant correlations ($|R|>0.7$) with the sulfate formation rate ($p < 0.01$, Figure S9)."

Table S4 Shapiro-Wilk normality test results for the analysis of correlation coefficients.

| Parameter | Statistic | p-value | Decision at level (5%) |
| --- | --- | --- | --- |
| $k_{SO4}$ | 0.90341 | 0.17551 | Can't reject normality |
| Chloride | 0.66506 | 3.96E-04 | Reject normality |
| Fe | 0.83678 | 0.02531 | Reject normality |
| Mn | 0.79415 | 0.00804 | Reject normality |
| Sulfate | 0.70692 | 9.83E-04 | Reject normality |
| Nitrate | 0.87657 | 0.07923 | Can't reject normality |
| CHO- | 0.80111 | 0.00964 | Reject normality |
| CHON- | 0.78144 | 0.00581 | Reject normality |
| Others- | 0.78595 | 0.00651 | Reject normality |
| CHO+ | 0.68063 | 5.52E-04 | Reject normality |
| CHON+ | 0.7749 | 0.00492 | Reject normality |
| CHN+ | 0.86772 | 0.06114 | Can't reject normality |
| Others+ | 0.88449 | 0.10007 | Can't reject normality |

[Figure]

* p<=0.01

Fig. S9 Heatmap of Spearman correlations between sulfate formation rate (kSO4) and other factors, including chloride, Fe, Mn, sulfate, nitrate, and different chemical species detected by ESI (-) and ESI (+) mode. Note that the calculations were based on the sulfate formation rate and the initial concentrations of the influencing factors in the bulk solution. The symbol * indicates significance, i.e., $p \leq 0.01$. Red color means positive correlation ($r > 0$) and blue color means negative correlation ($r < 0$). The darker the color, the higher the r value.

4.      Regarding Figure 5: This schematic may not be of broad interest and would be better replaced by a more good-looking representation, with the primary pathways retained.

**Response:**

We don't know how to interpret the comment on "good-looking". This is not a TOC artwork. We view that it is useful to include these important pathways. We slightly modified it to the following:

[Figure]

Figure 5. Conceptual diagram of PS and chloride mediated ROS and RCS production in the oxidation processes from S(IV) to S(VI)

5.      I would also suggest that the authors emphasize the atmospheric implications of their findings and provide more data on the relative importance of the observed photochemical events.

**Response:**

In our manuscript, we focus on comparing the photosensitizing abilities of different BB and NaCl chemical systems rather than quantifying their absolute values. The significant uncertainties in atmospheric aqueous PS concentrations ($2.3×10^{-13}$ to $1.6 ×10^{-10}$ M) (Wang et al., 2020), along with their varying photosensitizing capacities, would lead to significant uncertainties in calculating sulfate formation based on our findings. For example, we roughly estimated the sulfate formation rate under the scenario set by Cheng et al. (2016) and following the procedures proposed by Liu et al. (2021). Assuming an average PS molar mass of 170g $mol^{-1}$ and a pH of 6, the estimated sulfate formation rate ranged from $1.29×10^{-6}$ µg $m^{-3}$ $h^{-1}$ to 0.034 µg $m^{-3}$ $h^{-1}$, which spans several magnitudes. While this work serves as a foundational study for similar complex systems, further research, such as estimating PS* concentrations, is necessary for more accurate assessments.

6.       Finally, additional relevant literature on atmospheric photosensitization should be cited and discussed to help readers gain a deeper understanding of this interesting topic.

**Response:**

We have added additional relevant literature and discussions on atmospheric photosensitization into the manuscript (lines 43-49 and lines 310-318):

[revised manuscript text omitted]